# DNA methylation is reconfigured at the onset of reproduction in rice shoot apical meristem

Asuka Higo [1], Noriko Saihara[1,2], Fumihito Miura [3,4], Yoko Higashi[2], Megumi Yamada[2], Shojiro Tamaki[2], Tasuku Ito[5,6,16], Yoshiaki Tarutani[5], Tomoaki Sakamoto [7,12], Masayuki Fujiwara[7,13], Tetsuya Kurata[7,14], Yoichiro Fukao[7,15], Satoru Moritoh[8,17], Rie Terada[9], Toshinori Kinoshita [10], Takashi Ito [3], Tetsuji Kakutani[5,6,11], Ko Shimamoto[2] & Hiroyuki Tsuji [1]✉

DNA methylation is an epigenetic modification that specifies the basic state of pluripotent stem cells and regulates the developmental transition from stem cells to various cell types. In flowering plants, the shoot apical meristem (SAM) contains a pluripotent stem cell population which generates the aerial part of plants including the germ cells. Under appropriate conditions, the SAM undergoes a developmental transition from a leaf-forming vegetative SAM to an inflorescence- and flower-forming reproductive SAM. While SAM characteristics are largely altered in this transition, the complete picture of DNA methylation remains elusive. Here, by analyzing whole-genome DNA methylation of isolated rice SAMs in the vegetative and reproductive stages, we show that methylation at CHH sites is kept high, particularly at transposable elements (TEs), in the vegetative SAM relative to the differentiated leaf, and increases in the reproductive SAM via the RNA-dependent DNA methylation pathway. We also show that half of the TEs that were highly methylated in gametes had already undergone CHH hypermethylation in the SAM. Our results indicate that changes in DNA methylation begin in the SAM long before germ cell differentiation to protect the genome from harmful TEs.

[1] Kihara Institute for Biological Research, Yokohama City University, Yokohama, Kanagawa 244-0813, Japan. [2] Graduate School of Biological Sciences, Nara Institute of Science and Technology, Ikoma, Nara 630-0192, Japan. [3] Department of Biochemistry, Kyushu University Graduate School of Medical Sciences, Fukuoka, Fukuoka 812-8582, Japan. [4] Precursory Research for Embryonic Science and Technology (PRESTO), Japan Science and Technology Agency (JST), Saitama, Japan. [5] National Institute of Genetics, Mishima, Shizuoka 411-8540, Japan. [6] Department of Genetics, School of Life Science, The Graduate University for Advanced Studies (SOKENDAI), Mishima, Shizuoka 411-8540, Japan. [7] Plant Global Education Project, Graduate School of Biological Sciences, Nara Institute of Science and Technology, Ikoma, Nara, Japan. [8] National Institute for Physiological Sciences, Okazaki, Aichi 444-8585, Japan. [9] Graduate School of Agriculture, Meijo University, Nagoya 468-8502, Japan. [10] Institute of Transformative Bio-Molecules (WPI-ITbM), Nagoya University, Nagoya 464-8602, Japan. [11] Faculty of Science, The University of Tokyo, Bunkyo-ku, Tokyo 113-0033, Japan. [12] Present address: Faculty of Life Sciences, Kyoto Sangyo University, Motoyama, Kamigamo, Kita-Ku, Kyoto 603-8555, Japan. [13] Present address: YANMAR HOLDINGS Co. Ltd., Chayamachi 1-32, Kita-ku, Osaka 530-8311, Japan. [14] Present address: EditForce Inc., 4th Fl., Tenjin Fukuoka Seimei Bldg., Tenjin 1-9-17, Fukuoka 810-0001, Japan. [15] Present address: Graduate School of Life Science, Ritsumeikan University, Kusatsu, Shiga 525-8577, Japan. [16] Present address: Department of Cell and Developmental Biology, John Innes Centre, Norwich Research Park, Norwich NR4 7UH, UK. [17] Present address: College of Pharmaceutical Sciences, Ritsumeikan University, Kusatsu, Shiga 525-8577, Japan. ✉email: tsujih@yokohama-cu.ac.jp

DNA methylation is an epigenetic modification in which cytosine residues are methylated in three sequence contexts (CG, CHG, and CHH, where H is a nucleotide other than G). DNA methylation can affect multicellular development from pluripotent stem cells through gene regulation and transposable element (TE) silencing. In animals, DNA methylation specifies the basic state of pluripotent stem cells and regulates the developmental transition from stem cells to various cell types[1]. In flowering plants, the shoot apical meristem (SAM) contains a pluripotent stem cell population. The SAM is the origin of the aerial part of plants, including the germ cells, and plants begin to commit to sexual reproduction when the SAM is converted from a leaf-developing (vegetative) SAM to an inflorescence/flower-developing (reproductive) SAM[2,3].

Plant DNA methylation changes during many developmental processes including sexual reproduction[4,5]. The ways to change DNA methylation are categorized as reprogramming and reconfiguration. Reprogramming often includes two sequential changes in DNA methylation patterns: first, demethylation of DNA, and second, re-establishment of new DNA methylation patterns. Genome-wide reprogramming of DNA methylation is essential for proper embryonic development in animals[1,6]. In plants, DNA methylation reprogramming occurs in pollen, and particularly in TEs, imprinting genes, and some epialleles[7,8]. In addition, plant germ cell differentiation and fertilization results in changes in DNA methylation, and in most cases these changes are also considered as reprogramming; they involve hypermethylation in gametes and hypomethylation in their companion cells (central cells in the female and vegetative cells in the male)[4,5]. In contrast, the term reconfiguration includes genome-wide changes of DNA methylation patterns without the massive disappearance of DNA methylation[9]. CHH methylation is reconfigured throughout embryogenesis and germination, with methylation increasing during seed development and declining during germination[9–12].

The DNA methylome in the SAM and its dynamics at the vegetative-to-reproductive transition have so far eluded analysis because the SAM is situated deep in the vegetative organs, surrounded by leaf primordia, and is a minuscule tissue having a dome-like structure, about 50 μm in diameter in rice (Supplementary Fig. 1a). In this study, to reveal features of DNA methylation in the SAM and its mechanism of regulation, we generated single-base-resolution maps of cytosine methylation by whole-genome bisulfite sequencing (BS-seq), and also performed RNA-seq, small RNA-seq (smRNA-seq), and proteome analysis of the SAMs in the vegetative and reproductive stages and in mature leaf blades (Supplementary Fig. 1b and Supplementary Table 1). We found genome-wide CHH hypermethylation in the SAM and a further increase in CHH methylation at the onset of the reproductive stage, especially at the edges of TEs. We ascribed this hypermethylation to a protective function for the genome by silencing TE activity. We show that the RNA-dependent DNA methylation (RdDM) pathways contributes to CHH methylation in the SAM. We further propose that two rounds of DNA methylation occur during plant reproduction: first, reconfiguration at the edges of TEs in the SAM, and second, reprogramming in the bodies and at the edges of TEs during germ cell differentiation.

## Results

### CHH methylation globally increases in the reproductive SAM.
We undertook BS-seq with the post-bisulfite adapter-tagging (PBAT) method; this enables PCR-free library preparation from limited amounts of genomic DNA[13] for the SAM of rice, which can be cleanly and quickly isolated by hand dissection[2] (Supplementary Video 1). To generate whole-genome DNA methylome data for SAMs, we collected 164 vegetative and 140 reproductive SAMs and sequenced the nuclear genomic DNA to 24-fold and 25.2-fold coverage, respectively (Supplementary Table 1). To visualize the global DNA methylation pattern in each sample, average methylation rates in 1-Mb windows were plotted as a heat map on all 12 chromosomes of rice in all sequence contexts (Fig. 1a). For the CG and CHG contexts, methylated cytosines were enriched in the pericentromeric regions in vegetative and reproductive SAMs and mature leaves, and the methylation levels of these cytosines were generally comparable among the tested organs. Strikingly, however, the CHH context was globally changed: CHH methylation levels were higher in the vegetative SAM than in mature leaves, and global CHH hypermethylation occurred during the vegetative-to-reproductive transition in the SAM. To analyze the varied methylation between the organs, we made kernel density plots of methylation differences within 50-bp windows throughout the genome among the tested organs (Fig. 1b). The peak at zero in the kernel density plots showed that cytosines in CG were similarly methylated in the three organs. For CHG methylation, we observed a slight shift of the peak toward higher levels in the reproductive SAM (Fig. 1b). In contrast, for CHH methylation, we observed two peaks between the leaf and vegetative SAM: one corresponds to hypermethylation in the leaf (discussed later) and the other to hypermethylation in the vegetative SAM. We also observed a shift of the single CHH peak between the vegetative and reproductive SAM (Fig. 1b). These results show that CHH methylation is kept high in the vegetative SAM and globally increases in the reproductive SAM, with a slight increase in CHG methylation.

### CHH methylation is high in TEs in the SAM.
To examine DNA methylation patterns in different genomic features, we compared average methylation levels within gene bodies and TE bodies among the leaf, the vegetative SAM, and the reproductive SAM. CHH methylation rates were higher in the vegetative SAM than in the mature leaf, and CHH methylation increased in the reproductive SAM (Fig. 2a). To look more closely at the DNA methylation pattern along genes and TEs, we analyzed the DNA methylation profiles by metaplots. The profiles in the SAM were similar to the typical pattern previously reported in rice[14–16]; for genes, CG methylation is low at the edges of genes but high in the gene body, CHG methylation is low in gene bodies, and CHH methylation is high in the close vicinity of genes (Fig. 2b). For TEs, CG and CHG methylation are high in TE bodies, but CHH methylation is high at the edges of TEs (Fig. 2b). We then compared the level of DNA methylation among tested organs. The CG methylation profile was identical among the three tested organs; however, a slight difference was seen in CHG methylation levels between the SAM and mature leaves. Striking differences were found in CHH methylation levels among the tested organs: CHH methylation is lowest in the mature leaf, high in the vegetative SAM, and highest in the reproductive SAM, in the close vicinity of genes and TEs, in the regions upstream and downstream of TEs, and at the edges of TEs. To examine the relationship between TE size and methylation pattern, we categorized TE families into short and long TEs and analyzed methylation profiles in different families of TEs (Supplementary Fig. 2). We found CHH hypermethylation at the edges of short and long TEs. These results suggest that CHH methylation is higher around protein-coding genes and at the edges of TEs in the SAM than in the differentiated mature leaf, and that during the vegetative-to-reproductive transition the CHH methylation levels rise further in the SAM.

To characterize differentially methylated regions (DMRs) during the vegetative-to-reproductive transition in the SAM, we defined DMRs in each context by applying the same criteria as

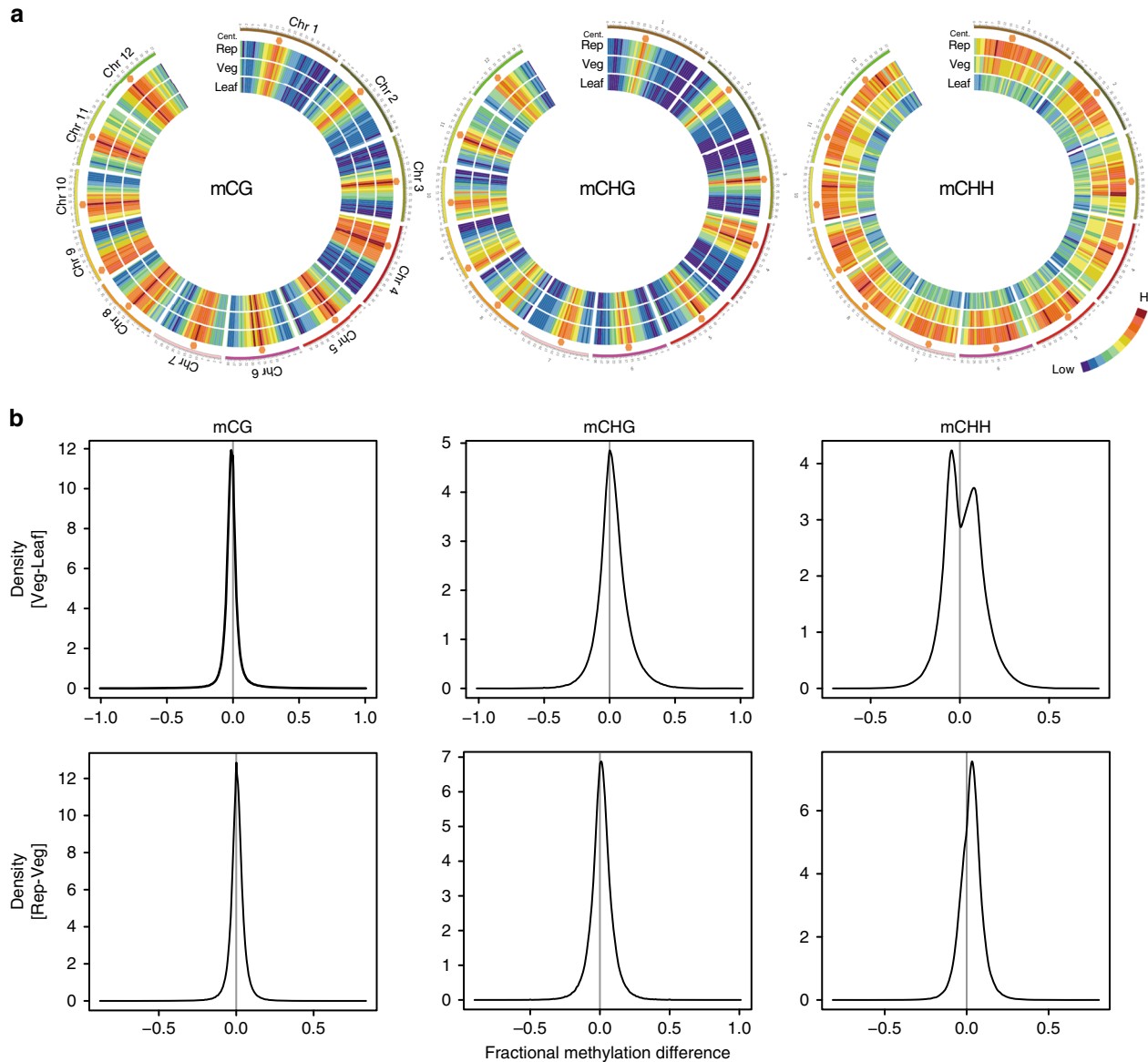

**Fig. 1 CHH methylation is kept globally high in the SAM and increases in the reproductive SAM. a** Heat maps of the 12 rice chromosomes show cytosine methylation levels in the vegetative SAM (Veg), reproductive SAM (Rep), and mature leaf blade (Leaf) for the CG (left), CHG (middle), and CHH (right) contexts. Average methylation rates were calculated within 1-Mb windows. The maximum for each context was set as the highest cytosine methylation rate among the three organs. Orange hexagons mark pericentromeric regions. **b** Density plots show the frequency distribution of the methylation differences within 50-bp windows of the whole rice genome between vegetative SAM and mature leaf blade (upper) or between reproductive SAM and vegetative SAM (lower) for CG (left), CHG (middle), and CHH (right). A shift of the peak away from zero represents a global difference.

Stroud et al.[17]. We detected DMRs in all contexts, and CHH DMRs were most abundant (Fig. 3a). Next, we examined the distribution of DMRs in genomic features such as genes, TEs, and intergenic regions. To elucidate DNA methylation dynamics during the vegetative-to-reproductive transition in the SAM, we focused on the Rep-hyperDMRs, which are hypermethylated in the reproductive SAM relative to the vegetative SAM. Among Rep-hyperDMRs, 71.5% overlapped with TEs, especially those in intergenic regions, while only 9.9% overlapped with protein-coding gene bodies (Fig. 3b). Of the TE-overlapping Rep-hyperDMRs, 41.1% corresponded to miniature inverted-repeat transposable element (MITE)-type TEs (Fig. 3c) in nucleotide length, although MITEs comprise only 7.0% of the total length of TEs in the rice genome (Table 1). In contrast, CHG methylation accumulates at longer TEs (Supplementary Fig. 3a, b). This suggests that MITEs are preferential targets of CHH

hypermethylation in the SAM. Both short and long TEs gain CHH methylation (Supplementary Fig. 2), although short MITEs are most highly enriched for CHH methylation (Fig. 3c). The reason for this difference may be that methylation rates vary according to TE length. CHH methylation rates are higher in short TEs (10–25% methylated at peaks) than in long TEs (5–18%) (Supplementary Fig. 2), resulting in effective DMR detection that leads to enrichment in short TEs, especially MITEs, among DMR-overlapping TEs (Fig. 3c). To examine the effect of MITE insertion on DNA methylation, we compared DNA methylation of genes with or without MITE insertions. MITEs preferentially occur near genes[14,18–23] (Supplementary Fig. 4a) and their presence or absence closely parallels the level of CHH methylation around genes (Supplementary Fig. 4b–e). This suggests that CHH methylation near genes reflects CHH methylation in MITEs near genes. To examine whether

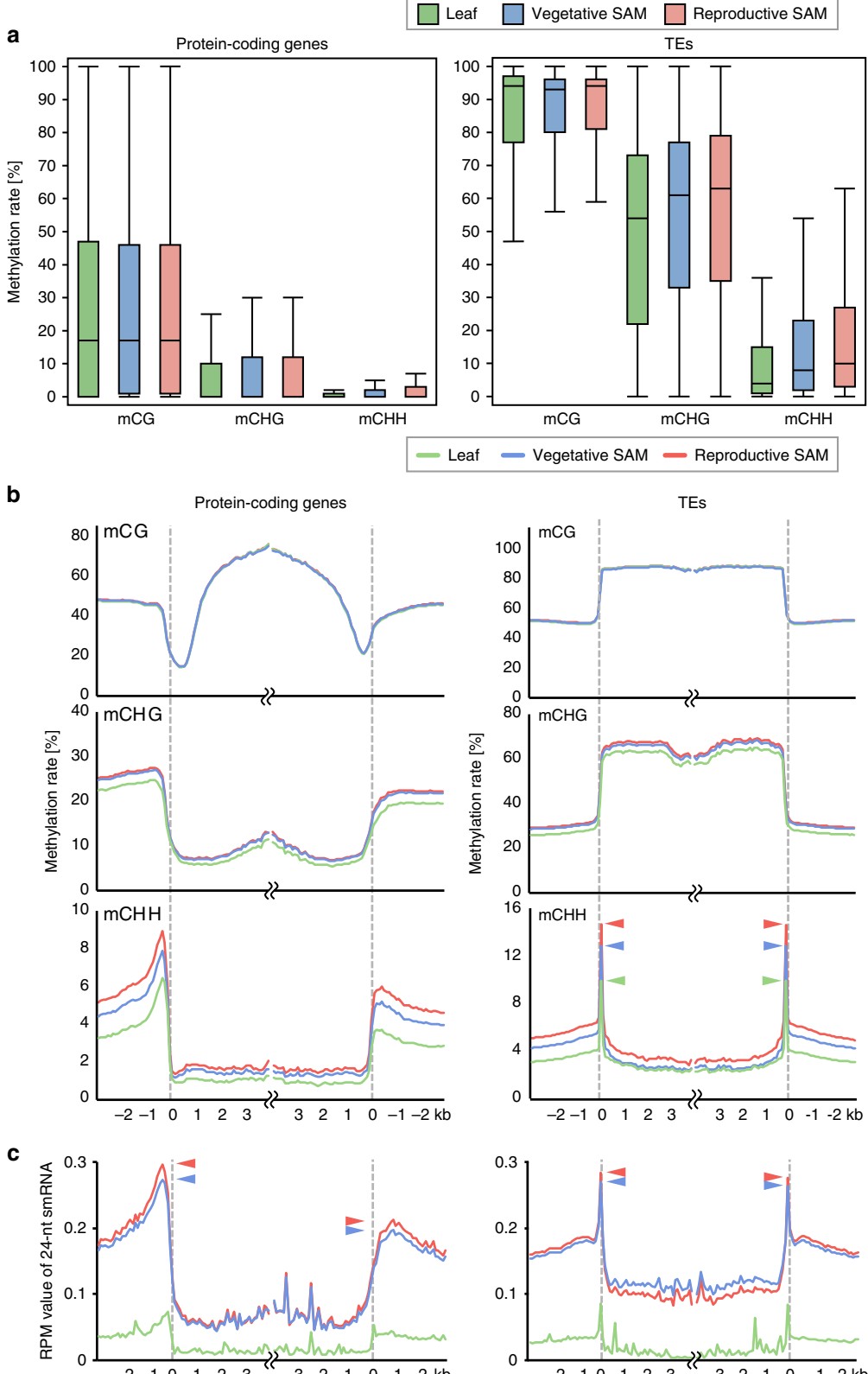

methylation profiles along TEs are affected by the DMRs, we made metaplots of TEs containing Rep-hyperDMRs. We found that the shapes of metaplots of TEs are not affected by the overlap with DMRs (Fig. 2b and Supplementary Fig. 3c).

Our analysis so far showed that CHH methylation is generally higher in the SAM than in the leaf. However, close inspection of

the kernel density plots (Fig. 1b) and DMR detection (Fig. 3a) revealed that there are regions that acquire CHH methylation in the leaf. To characterize these regions, we analyzed the distribution of DMRs hypermethylated in the leaf (Leaf-hyperDMRs) in genes and TEs. For Leaf-hyperDMRs, 70% overlapped with intergenic TEs (Fig. 3b, Leaf hyper); 41.6% of

**Fig. 2 CHH methylation level varies around both protein-coding genes and TEs. a** Box plots show cytosine methylation levels of bodies of protein-coding genes (left) and TE (right) in mCG, mCHG, and mCHH in mature leaf blade (green), vegetative SAM (blue), and reproductive SAM (red). **b** Metagene plots show patterns of DNA methylation for each context in mature leaf blade (green), vegetative SAM (blue), and reproductive SAM (red). Protein-coding genes (left) or TEs (right) were aligned at the 5′ end or the 3′ end and average methylation level for CG (top), CHG (middle), and CHH (bottom) contexts was plotted. Methylation level within each 100-bp window was averaged and plotted from 3 kb away from the protein-coding genes or TEs (negative numbers) to 4 kb into the annotated regions (positive numbers). Dashed lines represent the points of alignment. Arrowheads indicate the peaks at the edges of TE bodies. **c** Patterns of 24-nt small RNA expression in mature leaf blade (green), vegetative SAM (blue), and reproductive SAM (red) around protein-coding genes (left) and TEs (right). Protein-coding genes or TEs were aligned at the 5′ end or the 3′ end and average RPM values for smRNAs within 100-bp windows were plotted from 3 kb away from the protein-coding genes or TEs (negative numbers) to 4 kb into the annotated regions (positive numbers). Dashed lines represent the points of alignment. Arrowheads indicate the peaks around protein-coding genes and at the edges of TE bodies.

these TEs were MITEs (Fig. 3c, Leaf hyper). To examine whether methylation profiles along TEs are affected by the Leaf-hyperDMRs, we made metaplots of TEs containing Leaf-hyperDMRs, which indicated that the edges of TE bodies are hypermethylated in these TEs (Supplementary Fig. 3c), similar to the CHH methylation pattern in the SAM and other organs of rice. These results indicate that the CHH methylation profiles of leaf and SAM share similar features.

Our results suggest that CHH methylation generally declines during leaf differentiation from the vegetative SAM, but rises upon the vegetative-to-reproductive transition. To examine whether these two processes share DMRs, we analyzed the overlap between Leaf-hypoDMRs and Rep-hyperDMRs. Leaf-hypoDMRs represent regions hypomethylated in the leaf relative to the vegetative SAM, and thus reflect regions whose CHH methylation declines during leaf differentiation from the vegetative SAM. Rep-hyperDMRs represent regions hypermethy-lated in the reproductive SAM relative to the vegetative SAM, and thus reflect regions whose CHH methylation rises upon the vegetative-to-reproductive transition of the SAM. Only a part of these DMRs overlapped (Fig. 3d), suggesting that CHH methylation varies among different developmental processes.

To examine whether CHH methylation affects TEs globally or locally, we made density plots of the differences in CHH methylation between vegetative SAM vs. leaf and vegetative SAM vs. reproductive SAM for regions overlapping with MITEs (Supplementary Fig. 5). These plots showed a slight shift of the central peaks, indicating (i) hypermethylation of the vegetative SAM, in the vegetative SAM vs. leaf comparison, and (ii) hypermethylation of the reproductive SAM, in the vegetative SAM vs. reproductive SAM comparison. Because a peak shift in the density plots reflects a global change in methylation differences, this suggests that CHH methylation affects MITEs globally in different tissues.

To examine whether changes in CHH methylation affect CG and CHG methylation in the same regions, we analyzed CG and CHG methylation levels in CHH DMRs between the vegetative SAM and the reproductive SAM. This revealed that CG and CHG methylation rates were comparable between the vegetative and reproductive SAM (Fig. 3e), suggesting that changes in DNA methylation are independent in different sequence contexts in the SAM.

To examine whether increased CHH methylation contributes to TE silencing during the transition from vegetative to reproductive transition, we compared TEs that gain CHH methylation and those that are repressed in the reproductive SAM[2]. From our previously published data, we identified 8045 expressed TEs in the SAM; of these, 4263 gained CHH methylation in the reproductive SAM. In contrast, we identified 526 repressed TEs, of which 292 gained CHH methylation (Supplementary Fig. 6a). This suggests that repressed TEs were not necessarily enriched for CHH-hypermethylated TEs. We also analyzed the ratio of CHH methylation in the vegetative SAM

and reproductive SAM for expressed and repressed TEs by scatter plot (Supplementary Fig. 6b), which showed that both types of TEs gained CHH methylation similarly. These results suggest that DNA methylation increases globally in the reproductive SAM, possibly due to the activation of RdDM (discussed later). This global increase in CHH methylation may consolidate silencing of TEs that are not expressed even in the SAM, and/or be a prerequisite for repressing a subset of TEs prior to reproduction.

To further explore whether CHH methylation contributes to regulation of gene expression, we analyzed previously published transcriptome data[2]. All rice genes were categorized into three groups: genes whose expression is activated, repressed, or constitutively expressed in the SAM. We then plotted CHH methylation in the vegetative and reproductive SAM for the genes in each classification, divided into 1 kb upstream of the start of annotation, the gene body, and 1 kb downstream of the end of annotation (Supplementary Fig. 7). The results showed that CHH methylation in the reproductive SAM was higher than that in the vegetative SAM in the upstream and downstream regions of all groups. This suggests that CHH methylation is also globally up-regulated in genes during the transition from vegetative to reproductive SAM. We also found that genes whose expression is repressed are characterized by CHH methylation of the gene body already being high at the vegetative stage.

To examine whether accumulation of CHH methylation over MITEs in proximity to genes affects the expression of these nearby genes, we analyzed CHH methylation levels of MITEs in proximity to genes that are differentially expressed in the vegetative-to-reproductive transition in the SAM. We found that the changes in the gene expression levels did not correlate with changes in the CHH methylation levels (Supplementary Fig. 8), suggesting that CHH methylation does not have a strong effect on proximal gene expression in the SAM.

**RdDM contributes CHH hypermethylation in the SAM**. We next investigated the mechanism of CHH hypermethylation in the SAM. CHH methylation is regulated differently in euchro-matic regions and heterochromatic regions. For euchromatic regions, de novo methylation is induced by the RNA-directed DNA methylation (RdDM) pathways, which include 21-nt or 24-nt smRNAs and DOMAINS REARRANGED METHYL-TRANSFERASE 2 (DRM2; homologous to animal Dnmt3)[4,24,25]. For heterochromatic regions, CHROMOMETHYLASE 2 (CMT2) methylates cytosines in cooperation with the nucleosome remo-deler DECREASED DNA METHYLATION 1 (DDM1)[17,26]. All the methylation sites need to be re-methylated after every DNA replication to maintain the methylation rate. In asymmetric CHH sites, there is no symmetric hemimethylated double strand that can be used as a template for remethylation. The RdDM pathways are well-known mechanisms to methylate CHH sites at the edges of TEs, and CMT2 methylates CHH sites in the bodies of TEs[17] (Fig. 4a). To determine whether genes for CHH methylation pathways are expressed in the SAM, we conducted RNA-seq of

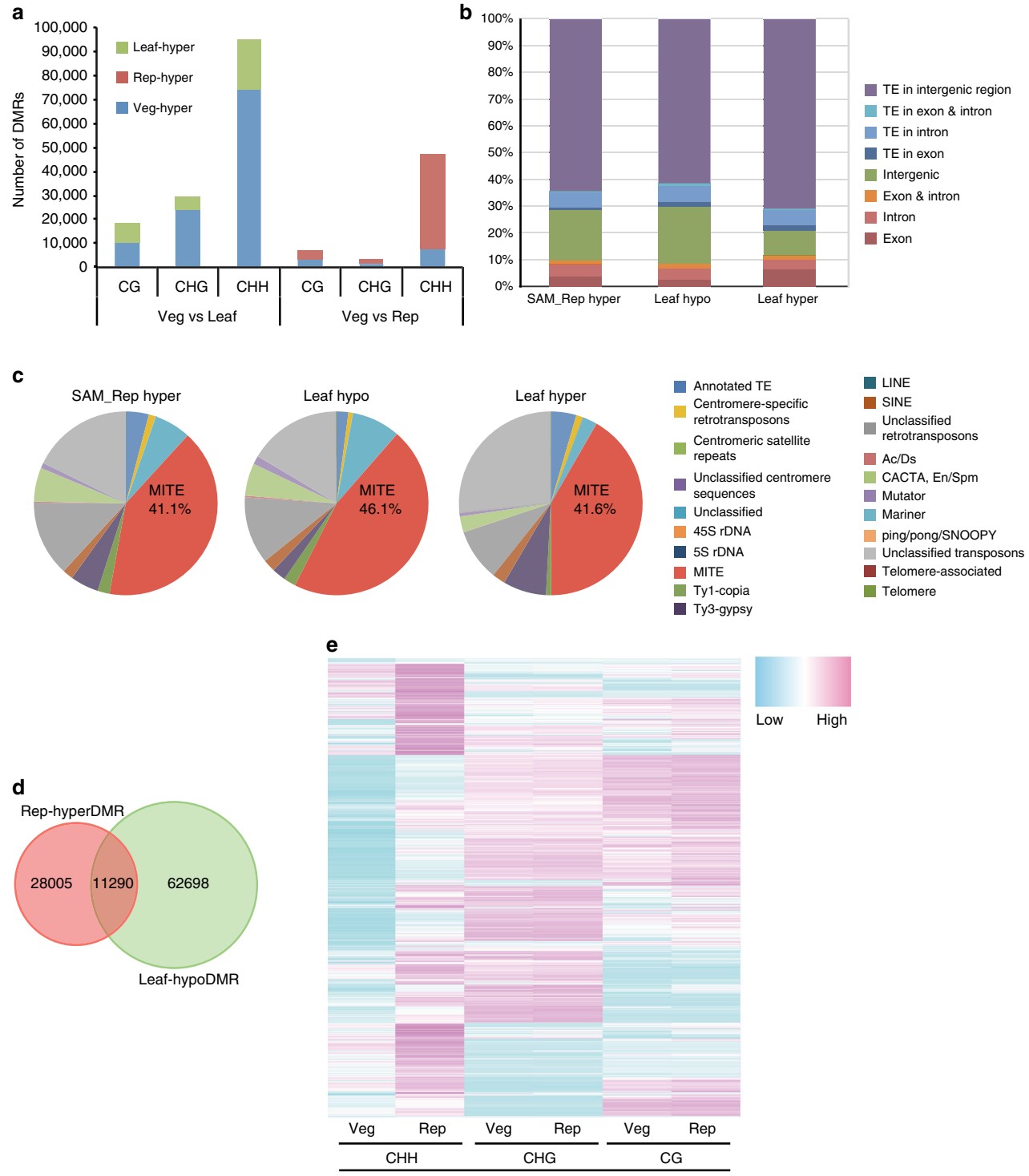

**Fig. 3 CHH hypermethylation occurs mainly on TEs. a** Numbers of DMRs between vegetative SAM and mature leaf (Veg vs. Leaf) and between vegetative SAM and reproductive SAM (Veg vs. Rep). Green, blue, and red indicate the portion of DMRs that are hypermethylated in mature leaf, vegetative SAM, and reproductive SAM, respectively. **b**, **c** Genomic features **b** and TE families **c** overlapping with CHH DMRs that are hypermethylated in reproductive SAM compared with vegetative SAM (left, SAM_Rep hyper), hypermethylated in vegetative SAM compared with mature leaf (middle, Leaf_hypo), and hypermethylated in mature leaf compared with vegetative SAM (right, Leaf_hyper). **d** Venn diagram showing the overlap of CHH DMRs between Rep-hyperDMRs and Leaf-hypoDMRs. **e** Heat map showing the methylation level for each context in the vegetative SAM (Veg) and reproductive SAM (Rep) for Rep-hyperDMRs.

**Table 1 Transposable elements overlapping with DMRs.**

| | Total length overlapping with SAM_rep hyper CHH DMRs | | Total length in genome | | Number of TEs |
|---|---|---|---|---|---|
| | bp | % | bp | % | |
| Others | 1,658,885 | 32.006 | 76,372,189 | 20.462 | 164,090 |
| MITE | 1,613,181 | 31.124 | 26,189,831 | 7.017 | 156,462 |
| CACTA, En/Spm | 333,344 | 6.431 | 11,880,018 | 3.183 | 18,055 |
| Ty3-gypsy | 271,641 | 5.241 | 31,876,324 | 8.540 | 24,844 |
| Annotated TEs | 234,481 | 4.524 | 54,427,496 | 14.582 | 16,937 |
| Ty1-copia | 106,136 | 2.048 | 5,170,109 | 1.385 | 8455 |
| SINE | 53,533 | 1.033 | 781,060 | 0.209 | 7587 |
| Mutator (MULE) | 39,017 | 0.753 | 804,829 | 0.216 | 3882 |
| Ac/Ds | 7088 | 0.137 | 328,847 | 0.088 | 1709 |
| Mariner (MLE) | 431 | 0.008 | 72,961 | 0.020 | 247 |
| ping/pong/SNOOPY | 197 | 0.004 | 126,318 | 0.034 | 169 |
| LINE | 148 | 0.003 | 189,264 | 0.051 | 531 |

the SAM. This dataset was combined with the RNA-seq dataset for the SAM and leaf from our previous publication[2]: replicate 1 in the leaf and replicates 1–3 in the reproductive SAM are from our previous publication[2] and other RNA-seq datasets are from our new experiments. We found that expression levels of canonical RdDM pathway components were generally higher in the SAM than in the leaf, but were comparable between the vegetative and reproductive SAM (Fig. 4b).

To examine the accumulation of smRNAs in the SAM, we performed smRNA-seq on the vegetative SAM, reproductive SAM, and leaf (Supplementary Table 1), which revealed that 24-nt smRNAs constituted 50.4% and 52.7% of expressed smRNAs in the vegetative and reproductive SAM, respectively, but only 9.0% in the leaf (Fig. 4c). This suggests that elevated levels of 24-nt smRNAs contributes to CHH hypermethylation in the vegetative and reproductive SAM to a greater extent than in the leaf; however, the accumulation of 24-nt smRNA per se cannot explain the increase in CHH methylation from vegetative SAM to reproductive SAM. We therefore speculated that smRNA profiles would differ in vegetative and reproductive SAMs. To determine the relationship between 24-nt smRNA profiles and CHH methylation profiles, we analyzed metaplots of 24-nt smRNAs along protein-coding genes and TEs in the vegetative SAM, reproductive SAM, and leaf. The reads per million mapped (RPM) values for 24-nt smRNAs were low in the leaf, high in the vegetative SAM, and higher still in the reproductive SAM in the vicinity of protein-coding genes and the edges of TEs, although there was only a small increase (Fig. 2c). In TE bodies, however, 24-nt smRNAs are less abundant in the reproductive SAM than in the vegetative SAM (Fig. 2c), perhaps because the transcription of small RNA precursors by RNA polymerase IV or their processing by RNA-dependent RNA polymerase2 (RDR2) and DICER-LIKE3 (DCL3) is slightly attenuated in regions corresponding to these long TE bodies. These results suggest that the subtle increase in 24-nt smRNAs makes a limited contribution to the increase in CHH methylation at the edge of TEs during the vegetative-to-reproductive transition in the SAM (Fig. 2b, c). To assess this more closely we analyzed smRNA profiles along the TEs overlapping with Rep-hyper DMRs and compared them with the smRNA profiles of the TEs not overlapping with those DMRs. This revealed that smRNAs were more abundant in TEs overlapping with Rep-hyper DMRs than in TEs not overlapping with the DMRs, whereas smRNAs were increased slightly in the reproductive SAM at the edges of TEs both overlapping and non-overlapping with the DMRs (Supplementary Fig. 9). Together, these results imply that 24-nt smRNA abundance in the SAM makes a limited contribution to the increase in CHH hyper-methylation in the reproductive SAM.

CHH methylation increased during the vegetative-to-reproductive transition in the SAM, but this increase was not coupled with changes in the mRNA expression level of RdDM pathway components between the vegetative and reproductive SAM (Fig. 4b). To gain insight into the regulation of CHH methylation in the SAM, we conducted a proteome analysis of the SAM (Fig. 4d, Supplementary Fig. 10a, and Supplementary Data 1) and found an increase in ARGONAUTE4 (AGO4) protein level, attributable to a post-transcriptional mechanism (Supplementary Fig. 10b). Our analysis of the SAM revealed that OsAGO4a and OsAGO4b accumulated to higher levels in the vegetative SAM than in mature leaves, and OsAGO4b increased in the reproductive SAM relative to vegetative SAM (Fig. 4d, Supplementary Data 1). The increase in OsAGO4b exceeds the overall changes of proteomes in the SAM (Supplementary Fig. 10a). This is the first report of post-transcriptional regulation of AGO4 family proteins. Because AGO1 protein level is known to be regulated by autophagy in the immune response[27], it is possible that OsAGO4b shares a similar mechanism to regulate protein abundance. Taking these observations together, the higher accumulation of OsAGO4b in the SAM may explain why CHH methylation changes dramatically relative to CG or CHG methylation in the SAM.

To assess the contribution of the RdDM pathways to CHH hypermethylation in the SAM, we analyzed whether SAM and leaf DMRs are hypomethylated in the *osdrm2* mutant relative to wild type (WT). To do this, we asked whether OsDRM2 contributes to hypermethylation in the SAM compared with the leaf by examining the overlap between Rep-hyper DMRs (regions hypermethylated in the reproductive SAM compared to the vegetative SAM), Leaf-hypo DMRs (regions hypermethylated in the SAM compared to the leaf), or Leaf-hyper DMRs (regions hypermethylated in the leaf compared to the vegetative SAM) and *osdrm2*-DMRs (regions methylated by OsDRM2). We calculated the methylation differences between WT and the *osdrm2* mutant[28] for SAM and leaf DMRs. Almost all Rep-hyper CHH DMRs, Leaf-hypo CHH DMRs, and Leaf-hyper CHH DMRs were hypomethylated in the *osdrm2* mutant relative to WT (Fig. 4e). These results suggest that the RdDM pathway keeps CHH methylation high in the SAM and increases the CHH methylation during the vegetative-to-reproductive transition in the SAM. RdDM pathways include a canonical pathway and a non-canonical pathway[25]. Because DRM2, AGO4, and 24-nt smRNA participate in both pathways, we could not determine which pathway(s) contributes to the hypermethylation in the SAM.

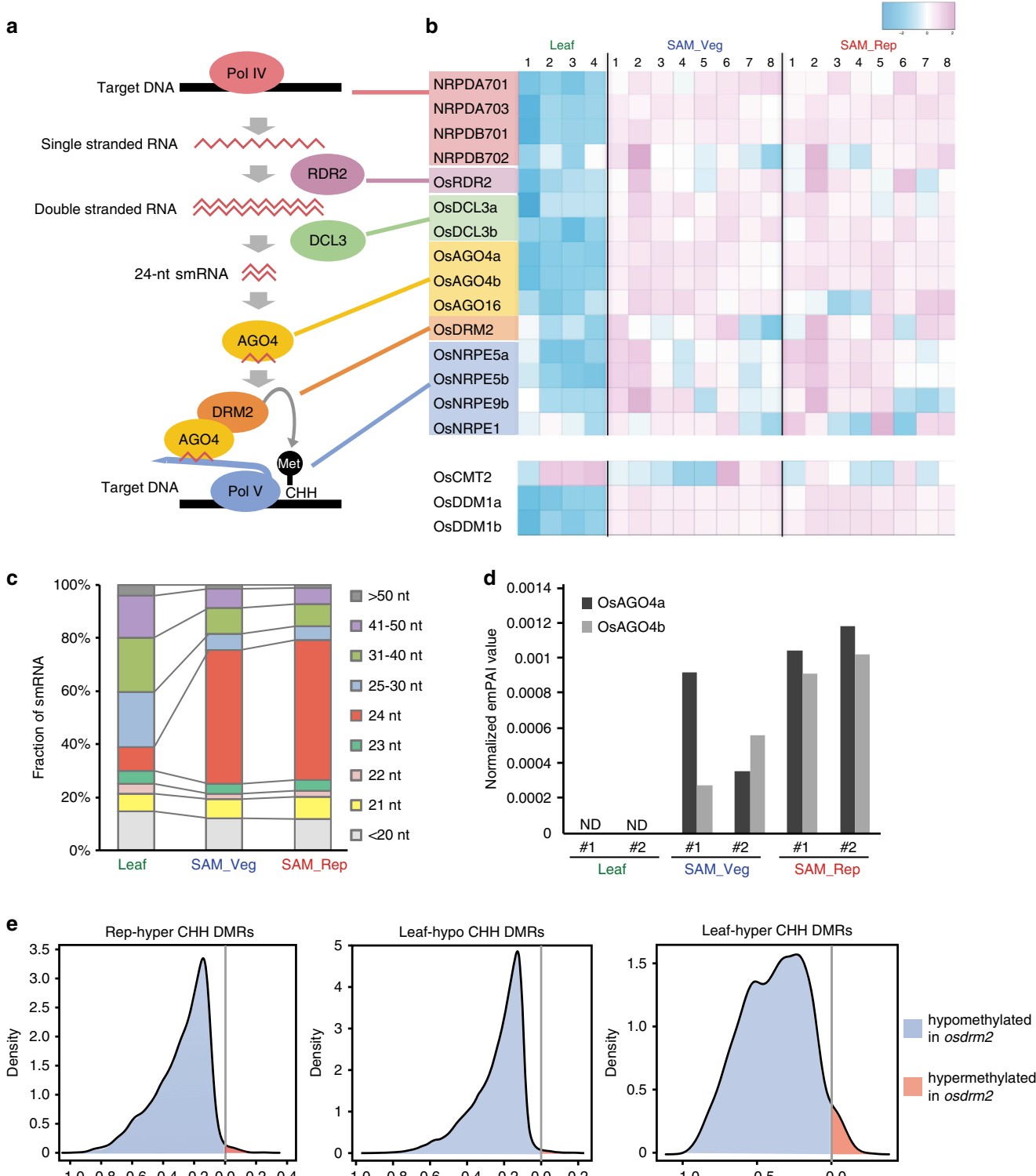

**Fig. 4 The RdDM pathway is active in the rice SAM. a** Schematic diagram of the canonical RdDM pathway. **b** Heat map showing the expression patterns of canonical RdDM pathway genes and rice homologs of *CMT2* and *DDM1* in mature leaf blade (Leaf), vegetative SAM (SAM_Veg), and reproductive SAM (SAM_Rep). Numbers above the heatmap show the number of samples: four, eight, and eight RNA-seq data sets were analyzed for mature leaf blade, vegetative SAM, and reproductive SAM, respectively. **c** Fraction of smRNAs for each size class relative to total smRNAs expressed in mature leaf blade (left, Leaf), vegetative SAM (middle, SAM_Veg), and reproductive SAM (right, SAM_Rep). The smRNAs were divided into nine classes according to their size. **d** Exponentially modified protein abundance index (emPAI)[41] values for OsAGO4a (dark gray) and OsAGO4b (light gray) from proteomic analysis. emPAI is quantitative information proportional to the protein concentration in the samples. ND not detected. #1 and #2 indicate each replicate of the proteomic analysis. **e** Density plots show the frequency distribution of CHH methylation differences within 50-bp windows of the Rep hyper CHH DMRs (left), Leaf hypo CHH DMRs (middle), and Leaf hyper CHH DMRs (right) between *osdrm2* knockout and WT leaf.

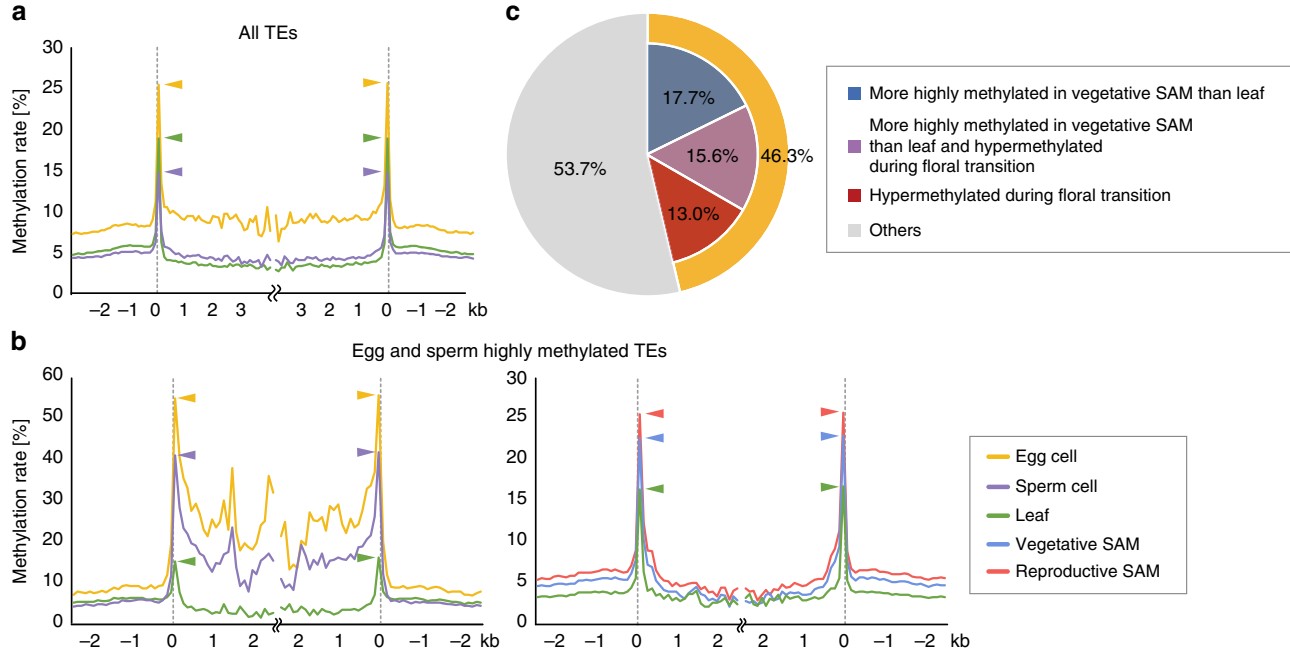

**Fig. 5 Many TEs highly methylated in germ cells undergo CHH hypermethylation in SAM. a** and **b** Metagene plots show patterns of DNA methylation for the CHH context in the egg cell (orange), sperm cell (purple), leaf (green), vegetative SAM (blue), and reproductive SAM (red) for all TEs **a** and TEs hypermethylated in the egg cell and sperm cell relative to the leaf **b**. Reported data for egg cell[16], sperm cell[31], and mature leaf[15] were re-analyzed using our methods. **c** Ratio of TEs that are hypermethylated in vegetative SAM compared with leaf (blue), hypermethylated in reproductive SAM compared with vegetative SAM (red), and hypermethylated in vegetative SAM and reproductive SAM compared with leaf (purple) to TEs hypermethylated in germ cells compared with leaves in the CHH context.

**TE CHH methylation in gametes are partly established in SAM.** Given the emerging evidence for DNA methylation reprogramming in plant germ cells[4,5,7,29,30], we examined whether CHH reconfiguration in the SAM is related to the DNA methylation reprogramming in germ cells. Reported DNA methylome data for egg cells[16] and sperm cells[31] of rice were re-analyzed with our methods and we confirmed that the TE bodies are CHH-hypermethylated in the egg cells (Fig. 5a). We then focused on the TEs that are highly methylated in both egg and sperm cells and examined whether they are hypermethylated in the SAM. Metaplots showed that the edges of these TEs are indeed hypermethylated in the vegetative SAM, and the methylation increased further at the edges of TEs in the reproductive SAM (Fig. 5b). In addition to the edges, the bodies of these TEs are further hypermethylated in germ cells (Fig. 5b). This suggests that the targets of CHH hypermethylation differ in the SAM and germ cells. From these observations, we hypothesized that a set of TEs are methylated in the SAM, and then further methylated in the germ cells. To test this, we asked whether TEs that are hypermethylated in germ cells include those that are hypermethylated in the SAM. Among the TEs highly methylated in both the egg cells and sperm cells for the CHH context, 46.3% had already undergone CHH hypermethylation in the SAM (Fig. 5c, $p < 2.2e-16$; Fisher's exact test).

To examine whether TEs are further silenced in the egg cell and sperm cell, we analyzed expression levels of TEs methylated in the SAM. We categorized expressed TEs into four groups: (1) those showing higher CHH methylation in reproductive SAM than in vegetative SAM, (2) those showing lower CHH methylation in reproductive SAM than in vegetative SAM, (3) those with similar CHH methylation in both SAMs, and (4) those without detectable CHH methylation in either SAM. We then compared expression levels of TEs in vegetative SAM, reproductive SAM, egg cell[32], and sperm cell[32] for each TE group

(Supplementary Fig. 11). We found no relationship between expression level and TE group, suggesting that TEs become silenced during the transition from vegetative to reproductive stages, independently of the change in CHH methylation and that SAM-expressed TEs are not further silenced in egg or sperm cells. Although expressed TEs were evidently not sufficiently silenced for their expression in the SAM to be prevented, the majority of TEs are already methylated and silenced in the SAM. Silencing of these TEs may be reinforced through an increase in CHH methylation in the reproductive SAM and in germ cells.

Interestingly, we found that TEs expressed and methylated in the SAM (group (1) and (2)) tend to be silenced in the egg cell, whereas TEs expressed but unmethylated in the SAM (group (4)) tend to be expressed, or to escape from silencing, in the egg cell (Supplementary Figs. 11 and 12). From this observation, we speculate that CHH methylation in the SAM may distinguish TEs that are to be silenced in the egg cell.

These results imply that a proportion of the TEs experience elevated levels of CHH methylation in the SAM that further increase in the germ cells.

## Discussion

In this study, we found that DNA methylation in the SAM is kept high in the CHH context and becomes higher during the vegetative-to-reproductive transition. Activation of RdDM pathway(s) through AGO4 regulation at the protein level may contribute to these methylation profiles. CHH methylation targets TEs in the SAM, suggesting that it functions to silence TEs upon vegetative-to-reproductive transition. Finally, we analyzed the relationship between TE methylation in the SAM and that in germ cells. A set of TEs undergo CHH hypermethylation in the SAM at their edges, followed by hypermethylation at TE edges and in TE bodies in the egg and sperm cells.

DNA methylation in the SAM is regulated through context-specific mechanisms. In our analysis, CHG methylation is almost unchanged, but CHH methylation increases, in the vegetative-to-reproductive transition of the SAM (Fig. 2b). A possible reason for this is the difference in activity between pathways responsible for CHH methylation and CHG methylation: RdDM is activated in the reproductive SAM through OsAGO4b protein accumulation (Fig. 4), but the mRNA level of *CMT3*, an enzyme responsible for CHG methylation in plants, is unchanged during the vegetative-to-reproductive transition in the SAM and CMT3 protein is below the detection limit of our proteome analysis. However, there may be a link between CHG methylation and CHH methylation because the *osdrm2* mutant shows a reduction of CHG methylation along with a reduction of CHH methylation. It has been observed that CHH methylation changes whereas CHG methylation is unchanged in the context of seed development and root cell-type specificity in *Arabidopsis*[9–12,33]. Further study is needed to clarify the context specificity of methylation regulation in the SAM.

TEs are highly expressed in the vegetative SAM, but some TEs are silenced during the vegetative-to-reproductive transition[2]. We asked whether this silencing correlates with the increase in CHH methylation in the reproductive SAM (Supplementary Fig. 6), and found that half of the expressed TEs are hypermethylated, and half of them are silenced. This suggests that silencing of expressed TEs does not necessarily coincide with CHH hypermethylation. We speculate that the function of CHH hypermethylation in the SAM is to reinforce pre-existing silencing of TEs. In the *Arabidopsis* SAM, TE silencing appears to be enforced via high expression of RdDM pathway components[34], and our findings thus illuminate a conserved feature in flowering plants.

We found that MITEs are the preferred target of CHH methylation in the SAM. The reason for this preference may be that MITEs are abundant in the rice genome and tend to insert near genes. TE silencing in the SAM may contribute to protecting the genome upon the onset of reproduction. To safeguard the genome against TE insertion, a reasonable strategy is to silence abundant TE families preferentially. MITEs are one of the most abundant TE families in rice and, because they tend to be inserted close to genes, their transcriptional activation may affect neighboring gene expression[35]. A tendency to insert near genes is also observed for *Arabidopsis* and human TEs[19–21]. For these reasons, silencing MITEs should be beneficial to protect the genome in the SAM. In addition to the silencing of TEs, CHH methylation at promoter regions can alter the gene expression level. MITEs prefer to insert into regions near genes such as promoters, and MITEs are highly methylated in the SAM, implying that they may often control gene expression in the SAM.

Our findings suggest a novel and earlier link between DNA methylation changes in the SAM and germ cells. A recent methylome analysis of the male germ cell lineage in *Arabidopsis* showed that the reprogramming of DNA methylation occurs before meiosis[30]. In the vegetative SAM, TEs are hypermethylated at their edges. This methylation is enforced in the vegetative-to-reproductive transition of the SAM. Germ cells differentiate in the reproductive SAM, and during this process many TEs are subjected to hypermethylation at their edges and in their bodies. Further hypermethylation in the CHH and CHG contexts in the germ cells may reflect a strict requirement for TE silencing in these cells. Such a two-step regulation may contribute to their attaining an epigenetic state suitable for them to function as germ cells.

Certain cells within the SAM may show a similar CHH methylation profile to the egg cells. They may represent the precursors of the germ line in the SAM. Germ cells develop from L2 of the SAM in *Arabidopsis*. Thus, it is possible that L2 cells express RdDM pathway genes to confer a CHH methylation pattern similar to that in in the germ cells, especially for egg cells: this is not so evident for the sperm cells because RdDM is diminished in pollen sperm cell[8]. If this is the case, L2 cells should express different levels of RdDM genes. To examine this, we analyzed the published layer-specific transcriptome data for *Arabidopsis* SAM cells[36] (Supplementary Fig. 13). This indicated that RdDM pathway genes showed no specific enrichment in any cell type, including L2. Thus, from the expression patterns of RdDM genes in L2, we could not conclude that L2 cell express a specific pattern of RdDM genes to establish a germ cell-specific methylation profile of the genome. In addition, unlike the *Arabidopsis* SAM, L2 cannot be clearly defined by cell-lineage analysis in the SAM of grass species including rice[37]. However, it is still possible that DNA methylation levels differ among cell types. Different cell types in the SAM show different cell proliferation activity, which may affect the cellular specificity of DNA methylation level: CHH methylation may well be more sensitive to cell proliferation activity, because it depends on the RdDM machinery and is not supported by symmetry-based maintenance. Further investigation will be required to reveal the cellular identity and contribution of subepidermal layer(s) of the SAM in grass species.

CHH hypermethylation of TEs in the SAM can be regulated through intercellular communication by smRNAs across specific cell types in the SAM. Such a mechanism to control TE expression is reminiscent of the role of PIWI-interacting RNAs in animals, which suppress TEs in the germ line[38,39]. In *Arabidopsis* pollen, transcriptional silencing has been reported for annotations that are targeted by smRNAs derived from the vegetative cell[7,8,29]. It is likely that these TEs are post-transcriptionally silenced. Whether hypermethylation by mobile smRNAs affects this silencing is debated. Hypermethylation in sperm cells may be caused by the depletion of linker histone H1 in *Arabidopsis*[40], and the contribution of RdDM may be limited because RdDM seems to be downregulated in the sperm cell[8]. However, Kim et al.[31] found that the loss of active DNA demethylation in pollen vegetative cells leads to a decrease of CHH methylation in sperm cells in rice. In the *Arabidopsis* root apical meristem, excess 24-nt smRNAs are thought to be produced in the columella and transported to the nearby stem cells to reinforce silencing of TEs[33]. These insights and our present findings emphasize the importance of epigenetic regulation via smRNAs for reproduction in eukaryotes.

## Methods

**Plant materials and growth conditions.** The Japonica rice cultivar Norin 8 (N8) was used for SAM analysis. Plants were grown in climate chambers at 70% humidity under short day conditions with daily cycles of 10 h of light at 27 °C and 14 h of dark at 25 °C. Light was provided by fluorescent white light tubes (400–700 nm, 100 μmol m$^{-2}$ s$^{-1}$). Vegetative and reproductive SAMs were isolated by hand dissection of basal region of rice under microscopy[2] (Supplementary Fig. 1, Supplementary Video 1). Whole leaf blades of 25-day-old plants were used as mature leaf blades.

**Resequencing of N8 rice cultivar.** Total DNA was extracted from mature leaf blades of 25-day-old N8 seedlings using a DNeasy Plant Mini Kit (Qiagen). Sequencing was performed using HiSeq2000 (Illumina). Reads were mapped to Osa_RGAP ver7 from cultivar Nipponbare using BWA[42], and SNPs and short insertions or deletions were detected with SAMtools version 0.1.18[43,44]. To prepare the N8 genome sequence for methylome analysis, Osa_RGAP ver7 was corrected at each SNP but insertions or deletions were not corrected so as to maintain the positions of genes and TEs in the genome annotation to facilitate data analysis.

**Bisulfite-seq and data analysis.** For genome-wide bisulfite sequencing of the vegetative and reproductive SAMs, genomic DNA was extracted from 164 vegetative SAMs and 140 reproductive SAMs. About 50 SAMs were sampled in 50 μl of Buffer AP1 of the DNeasy Plant Mini Kit (Qiagen), heated at 65 °C for 10 min with gentle mixing. The SAM suspension was then mixed with 16 μl of Buffer AP2 and placed on ice for 5 min. After centrifugation at 20,000×*g* for 15 min at 4 °C, the

supernatant was transferred to a new tube; 5 µl of 10× PCR buffer from Ex Taq (Takara Bio) and 5 µl of RNase One (Qiagen) were added, and the solution was incubated for 30 min at 37 °C. The reaction mixture was then supplemented with 1 µl of Protease K (Qiagen) and incubated for 10 min at 50 °C to obtain the DNA extract. DNA extraction was repeated three times with two other groups of 50 or 60 SAMs. To obtain DNAs from a total of about 150 SAMs, the three DNA extraction mixtures were combined into a single tube. DNA was purified using Agencourt AMPure XP (Beckman). Illumina Sequencing libraries (100 bp single-end) were constructed using the PBAT method[13] and sequenced with Illumina Hiseq 2000. Reads were mapped to the N8 genome using Bismark[45].

Circos software[46] was used to construct Circos plots. Kernel density plots were generated by comparing the average cytosine methylation rate within a 100-bp window between vegetative SAM and either mature leaf blades or reproductive SAM. Windows with at least four cytosines that were each covered by at least four reads in at least one sample were used. DMRs were identified using the same strategy as Stroud et al. (2013)[17]. Briefly, the genome was tiled into 100-bp windows within which the number of called Cs and Ts were compared across samples. Windows with at least four cytosines that were each covered by at least four reads in at least one sample, absolute methylation difference of 0.4, 0.2, and 0.1 for CG, CHG, and CHH, respectively, and Benjamini–Hochberg corrected FDR < 0.01 (Fisher's exact test) were selected. DMRs within 200 bp of each other were merged. TE sequences were downloaded from the *Oryza* Repeat Database (http://rice.plantbiology.msu.edu/annotation_oryza.shtml).

**RNA-seq and data analysis**. Total RNA isolation from vegetative or reproductive SAM and mature leaf blades, Illumina-sequencing library (100 bp single-end) construction. The reads were aligned to the rice genome using TopHat 2.0.4 with default parameters. After normalization, differentially expressed genes were extracted with multiple comparison correction. False discovery rate of <0.05 was chosen as the cutoff for determining whether differential gene expression was significant.

**smRNA-seq and data analysis**. Total RNA was isolated from vegetative SAM, reproductive SAM, and mature leaf blades as described previously[2]. Small RNAs shorter than 100 bp were extracted from the gel after separation by electrophoresis and sequenced with the Illumina platform.

**Proteome analysis**. *Sampling*: We collected 10 SAMs from each of 10 different plants for each replicate of the analysis. We also sampled and mixed three leaves from each of three different plants for each replicate.

*Protein extraction and digestion*: We extracted proteins from the samples with Laemmli sample buffer and incubated them at 65 °C for 15 min, and then subjected them to SDS–PAGE [acrylamide concentration 10.5% (w/v)] (Supplementary Fig. 14). After staining by Flamingo (BioRad, Hercules, CA, USA), we sliced each lane of the gel into four parts of equal length. We washed the sliced pieces twice with HPLC-grade water with 30% (v/v) acetonitrile (Kanto Chemical, Tokyo, Japan), once with 100% acetonitrile, and then dried them in a vacuum concentrator. We treated the dried gel pieces with 2 µl of 0.5 µg µl$^{-1}$ trypsin (sequence grade; Promega, Madison, WI, USA) in 50 mM ammonium bicarbonate[47], followed by incubation at 37 °C for 16 h. We recovered the digested peptides from the gel pieces by extracting twice with 20 µl of 5% (v/v) formic acid/ 50% (v/v) acetonitrile. The peptides were combined and dried in a vacuum concentrator.

*LC–MS/MS analys*: We performed LC–MS/MS analyses with an LTQ-Orbitrap XL-HTC-PAL-Paradigm MS4 system. Peptides were digested with trypsin and then loaded on the column (75 µm internal diameter, 15 cm; L-Column, CERI, Auburn, CA, USA). At this step we used a Paradigm MS4 HPLC pump (Michrom BioResources) and an HTC-PAL autosampler (CTC analytics, Zwingen, Switzerland). We prepared two buffers, 2% (v/v) acetonitrile and 0.1% (v/v) acetic acid (buffer A), and 90% (v/v) acetonitrile and 0.1% (v/v) acetic acid (buffer B). We applied a linear gradient from 5% to 45% by mixing buffers A and B, for 25 min, and after elution from the column we introduced the eluted peptides into an LTQ-Orbitrap mass spectrometer (Thermo Fisher Scientific, Bremen, Germany). The flow rate was 300 nl min$^{-1}$ and the spray voltage 2.0 kV. We set the range of MS scaning as *m/z* 200–2000, and the top three peaks were subjected to MS/MS analysis.

*Conditions of database search*: We compared the spectra with the protein database OSMSU (http://rice.plantbiology.msu.edu) using the MASCOT server (version 2.4.1, Matrix Science, London, UK). We used the following parameters for the MASCOT search: set off the threshold at 0.05 in the ion score cut-off, peptide tolerance at 10 ppm, MS/MS tolerance at ±0.5 Da, peptide charge of 2+ or 3+, trypsin as enzyme allowing up to one missed cleavage, carbamidomethylation on cysteines as a fixed modification and oxidation on methionine as a variable modification. Details of protein identification are presented in Supplementary Data 2.

*Calculation of normalized emPAI values*: We calculated the "normalized emPAI value" in Fig. 4d by dividing emPAI[41] of each protein by the sum of emPAI values of all the proteins in each sample[47]. This normalization was applied to compare between different samples with different sums of emPAI values.

**Reporting summary**. Further information on research design is available in the Nature Research Reporting Summary linked to this article.

## Data availability

The accession number for the N8 genome, the BS-seq data, the RNA-seq data, and the smRNA-seq data reported in this paper are DRA007588. Proteomic data associated with this study have been deposited in PRIDE under accession number PXD020608. Source data underlying Fig. 5c and Supplementary Fig. 1b are provided as a Source data file. Source data are provided with this paper.

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

## Acknowledgements
We thank Taiji Kawakatsu, Hisato Kobayashi, and Tetsu Kinoshita for comments on this work, and Ian Smith for critical reading of the manuscript. This study was supported by MEXT KAKENHI, Grants-in-Aid for Scientific Research on Innovative Areas (numbers 16H06464 and 16H06466 to H.T. and 15H05956 to T. Kinoshita), a Grant-in-Aid for Scientific Research (A), number 16H02532, to H.T., and by Core Research for Evolutionary Science and Technology (CREST) of the Japan Science and Technology Agency (JST) JPMJCR16O4 to H.T.

## Author contributions
H.T. and K.S. conceived and designed the study. N.S., Y.H., and M.Y. sampled the SAMs, and acquired and analyzed data. F.M. and T.I. conducted DNA methylome library preparation by the PBAT method. Y.H., S.T., T.S., and T. Kurata acquired and analyzed the mRNA-seq data. M.Y., M.F., and Y.F. acquired and analyzed the proteome data. N.S., R.T., T.I., Y.T., and T. Kakutani analyzed DNA methylome data for osdrm2 knockout rice. T. Kinoshita supported establishment of data analysis pipelines. A.H. and H.T. analyzed data and drafted the manuscript.

## Competing interests
The authors declare no competing interests.
