## [Peer Review File · Nature Communications]

Reviewers' comments:

Reviewer #1 (Remarks to the Author):

The manuscript "DNA methylation is reconfigured at the onset of the reproductive phase in the rice shoot apical meristem" by Higo and co-workers describes the genome-wide DNA methylation pattern of the shoot apical meristem (SAM) before and after the transition to the reproductive phase. The SAM contains the stem cells and gives rise to the germline in flowering plants and hence the epigenetic state of this tissue is of special interest. Using whole genome sequencing of bisulfite-converted genomic DNA (WGBS), the authors observe that it is mainly the DNA methylation pattern in the asymmetric CHH context that shows the highest dynamics in the meristem, with higher levels compared to leaf tissue and increasing methylation during the reproductive transition. This is in accordance with previously published transcriptome data in *Arabidopsis thaliana* and the authors also describe high levels of transcripts involved in establishing CHH methylation in the SAM by analysing publically available data in rice. These transcriptomes point towards a role of the RNA directed DNA methylation (RdDM) pathway to be highly active in the SAM. However, when looking into the WGBS of a mutant in this pathway(s), there is only little overlap between RdDM-dependent CHH methylation and regions that show high CHH methylation in the meristem compared to leaf tissue or during the transition to reproduction within the SAM. Higo and co-workers chose to use leaf tissue of a RdDM mutant instead of SAM material, which makes it difficult to conclude which DNA methylation pathways are responsible for the DNA methylation dynamics in the SAM in rice.

Moreover, the SAM contains several specialized tissues with diverse functions, such as a quiescent center-like cells as well as a proliferating stem cell population. One can expect that these cells contain a different epigenome, according to their function and distinct roles in providing the future germline, which cannot be revealed when looking at a whole SAM methylome. A recent report described the transcriptome and chromatin H3K4me3 and H3K27me3 profiling of specific meristematic cells during the transition to flowering using an established cell type-specific nuclear isolation protocol (You et al, Nature Communications 2017). Such cell type-specific methylome profiling in gametophytes and seeds in rice have been published, revealing clear differences with implications for the work presented here. Yet, CHH-hypermethylated regions in the egg cell seem largely independent of those that gain methylation in the SAM during the transition to reproduction, questioning the functional role of a putative germline in the SAM. Taken together it is difficult to obtain a clear picture from the current work.

Further remarks are listed below:

1) The authors state several times that the germline is not clearly defined in plants and that this is the main reason for the lack of knowledge about DNA methylation reprogramming. Yet, the authors' work does not contribute to solve this problem. Therefore the respective parts should be re-written, i.e. focus on meristematic development without focusing on the germline that cannot be tackled properly with the current results.

2) The term 'reprogramming' is largely avoided in expense of 'reconfiguration', which is not clearly defined either. Therefore the introduction of a novel term does not help. In most cases 'reprogramming' has been used to explain different aspects of DNA methylation dynamics, including qualitative and quantitative differences, but a unambiguous definition of the term is still missing. It would be helpful to give a brief definition how the authors define 'reconfiguration' in the context of DNA methylation.

3) The Kernel density plot shows both, gain and loss of CHH methylation when comparing leaf and vegetative SAM tissues (Figure 1b). However, throughout the manuscript only CHH hypermethylation in the SAM is investigated. What about the regions showing CHH hypomethylation in this comparison, especially given that it is very prominent? Also, the meta-gene and meta-TE methylation profiles might be more informative for those TEs containing DMRs

(hypo- and hyper-CHH DMRs) in addition to presenting all gene and TE annotations.

4) There have been four publications describing the DNA methylation dynamics during embryogenesis, yet only one is cited here. Please also cite the other three articles (Bouyer et al, *Genome Biol* 2017; Lin et al, *PNAS* 2017; Narsai et al, *Genome Biol* 2017).

5) Controls are missing that monitor the transition from vegetative to reproductive state in the SAM. Marker genes specific for each phase should be tested in RT-PCR experiments on the material used for WGBS.

6) In the analysis of DMRs affecting TEs I was missing a genome-wide profile of the different TE families/classes (Figure 3d). In the current form, one cannot conclude if a certain TE family, such as MITEs, are enriched, or if they mainly reflect the global average of TEs as present in the genome.

7) The authors state that the MITEs are located close to genes, reflecting the DNA methylation pattern seen for genes (line 120-122). This could and should be shown as a graph in the figure.

8) It is surprising that leaf vs. vegetative SAM DMRs do not overlap with vegetative vs. reproductive SAM DMRs (Figure 3e). It would be interesting to see the global DNA methylation pattern, i.e. to which extent does the dynamic of CHH methylation affect TEs globally or only specifically? Also, why do the authors analyse the vegetative vs. reproductive SAM DMRs with respect to their association with TEs, but not the leaf vs. vegetative SAM CHH DMRs? This should be completed.

9) The authors state that RdDM methylates CHH in TE bodies (line 135-136). This is misleading as the majority of RdDM-dependent CHH methylation takes place at TE borders and the TE bodies are mainly CHH methylated by CMT2 (Stroud et al, *Nature Struc. Mol Biol* 2014). However, small TEs are often mainly targeted by RdDM, thus covering also the TE body as a consequence. This needs to be clarified in the manuscript.

10) Given that DRM2-dependent CHH methylated regions and leaf vs. vegetative SAM- as well as vegetative vs. reproductive SAM-DMRs do not overlap much (Figure 4f), the authors should also analyse cmt2 mutants. Moreover, instead of using leaves, the SAM should be used as material.

11) The strong contribution of sRNAs in the leaf with >25nt size is suggested to result from PolIV primary transcripts (line 142-144 and Figure 4c). However, transcript levels of PolIV subunits are low in leaf tissue, but very high in the SAM, questioning the given explanation for this observation.

12) In line 155-158 the authors cite a publication, suggesting mobility of 24nt sRNAs from surrounding tissue to reinforce TE silencing, which was shown for 21-22nt sRNAs in pollen, but not for sRNAs in the ovule or seed as mentioned in the manuscript. This needs to be corrected.

13) The sRNA profile between vegetative and reproductive SAM does not change much, which is in agreement with the stable expression patterns of the RdDM components. How does the sRNA distribution look like for those sRNAs that target DMRs detected in the SAM during phase transition?

14) The authors mention a SAM proteome, but the only proteins shown are OsAGO4a and OsAGO4b. Therefore, it is unclear how these two proteins behave in comparison with the overall pattern. For instance, it was suggested that translational control might be involved in the transition from vegetative to reproductive development in the SAM (You et al, *Nature comm.* 2017). The authors need to provide further information on this aspect.

15) In line 178-180 the authors propose another de novo methylation pathway beside RdDM to

contribute to elevated CHH methylation levels in the SAM. It should be taken into account that there are non-canonical RdDM pathways that have been studied in more detail in Arabidopsis, but all rely on sRNAs (see Cuerda-Gil & Slotkin, Nature Plants 2016 for example).

16) The DNA methylation profile shows higher levels in the egg cell compared to all other cells or tissues investigated so far, including the SAM. This, together with the observation that CHH hypermethylation in this cell does not correspond to DNA methylation dynamics in the SAM, challenges the reprogramming concept and the order of events. Moreover, it questions the approach to use the entire SAM in order to reveal specific DNA methylome reprogramming. The authors state that the CHH methylation profile is qualitatively different in egg cells and in the SAM, yet the sRNA profile in the SAM matches the CHH methylation profile in the egg cell. Therefore it is possible that certain cells within the SAM show a similar CHH methylation profile compared to the egg cells, and others maintain CHH methylation mainly at the TE borders.

Reviewer #2 (Remarks to the Author):

Higo et al show that the reproductive shoot apical meristem (SAM) of rice is genome-wide hypermethylated for the CHH context in comparison to the vegetative SAM. Furthermore, their results indicate that the methylation patterns are modified in some genomic regions between vegetative and reproductive SAM. Therefore, this process is not only a hypermethylation of the genome, but also a structural change. So far, genome-wide DNA methylation is described for female and male germ cell formation and for developing seeds, but our knowledge remained elusive for the progenitor cells of germ cells in regard to changing DNA methylation.

The authors also provide a deeper insight for the mentioned elevated CHH hypermethylation by showing that both transposable elements (TE) and protein-coding genes are affected. They discover that MITE transposons are more often hypermethylated than other TEs in reproductive SAM and they suggest that the genome is protected by this mechanism. This suggests that the MITEs may be the major threat during sexual reproduction in rice, possibly because MITEs are often in vicinity to protein-coding genes. The role of MITEs should be analysed in more detail in the manuscript, by analyzing DNA methylation mutants and activation of MITEs; and/or in another rice accession with higher or lower amounts of MITEs and DNA methylation analyses.

The authors show that egg cells have an even higher level of CHG and CHH DNA methylation than the reproductive SAM, which will give rise to the egg cells during the plant's life cycle. That indicates that the change between vegetative and reproductive SAM is characterized by a change mostly in CHH methylation while the maturation of egg cells involve an even higher DNA methylation in the CHH and the CHG context. However, the biological reason for those changes remains elusive. Why do the progenitor cells of the germ cells (in the SAM) are not protected from TE activity during all stages of development? This should be discussed.

The activity of the RNA dependent DNA methylation (RdDM) can be partially responsible for the hypermethylation between vegetative and reproductive SAM according to the authors. They performed smRNA-seq and proteome analysis. The results of Higo et al. show that 24nt smRNA are slightly enriched in reproductive SAM in comparison to the vegetative SAM. However, Figure 4d shows that 24nt smRNA is less abundant in the TE bodies in reproductive SAM compared to vegetative SAM. This should be discussed.

Furthermore, the authors discover that AGO4 is higher expressed in the reproductive SAM. However, the methylation sites catalyzed by the AGO4 associated DRM2 DNA de novo methyltransferase are just partially overlapping with the hypermethylated sites in the reproductive SAM. The authors suggest that DDM1 which is higher expressed in the SAM tissue also plays a role for the increased hypermethylation.

In the end, the authors could not completely explain the mechanism of the hypermethylation. In my point of view, this should be addressed/discussed further.

Additional points:

Line 123: This hypothesis should be further discussed: Are there other TEs that are close to genes? Do these have the same distribution of CHH methylation than MITEs? Are there similar observations known in Arabidopsis or even animals?

Line 124: The reason why Rep-hyperDMRs and Leaf-hypoDMR are compared should be explained more: figure 1a shows high CHH methylation areas of Rep but the hypermethylated areas of leaf do not overlap very often.

Line 166: It is unclear what is referred to: certain cell types of the SAM? Please explain.

Line 178: I think it would be better to explain why leaf-hypoDMRs and osdrm2-hypoDMR are compared. The de novo DNA methyltransferase DRM2 is unlikely the reason for the hypomethylation in leaf, but it may be for the hypermethylation in Reproductive SAM.

Suppl.page3 Line 18: Where are those methods described?

Reviewer #3 (Remarks to the Author):

A dedicated germ line that is established early during embryonic development produces gametes in animal development. However, in plant development, the gamete forming cells are thought to arise from the L2 layer of SAMs. Authors analyze CHH methylation status in both vegetative and reproductive SAMs, which was found to be higher. Authors combine transcriptome and proteome data sets to show that the higher CHH methylation requires the activity of RNA-dependent DNA methylation pathway. They further found that DNA methylation patterns at transposons in the SAM partially overlapped with methylation patterns observed in reprogramming egg cells. Based on these observations, authors argue that reconfiguration of DNA methylation precedes germ cell differentiation in SAMs. Authors data showing higher levels of CHH methylation very early in vegetative SAMs in showing that reconfiguration of the methylation occurs very early during development is interesting. The following points must be clarified before linking observed methylation changes in SAMs to germ cells.

Comments

1. It is well documented that germ cells are specified from L2-layer derived cells in SAMs. Since authors use the whole SAMs for their analysis, it is not clear whether the observed methylation patterns are layer specific or occurs in all cell layers. What if the observed methylation changes occurred in all cell layers except in the L2 layer? I understand that it may be technically difficult to isolate cells from different layers in rice SAMs. Therefore, authors must be careful in interpreting these results and bring this point into their discussion. The layer specific transcriptome for Arabidopsis SAM cells is available (Yadav et al., Development 2014, 141(13):2735-44; Yadav et al., PNAS 2009, 106(12):4941-6). Authors must examine these data sets to see whether the RNA-dependent DNA methylation pathway components are enriched in any specific cell layers. I understand that the regulatory RNAs can move across cells, which can result in non-cell autonomous effects, but efforts should be made to see whether the pathways regulating methylation are layer-specific.
2. Line 50, misspelling of 'begins'
3. Line 92, Fig 1b It looks like there is a shift as well for CHG.
4. Line 97, Authors use the abbreviation TE (I assume transposable element) but I don't see it explicitly defined within the paper. It might make it more clear to use it together with the full name first.
5. Line 87, Do authors have any theories why the CHH context was changed so dramatically relative to the CG or CHG contexts between the conditions examined? What is the possible underlying differences in functionality between these different types of methylation from previous studies that could explain this?
6. Line 104, For readers that do not follow methylation field closely, authors should consider elaborating on the biological significance of the differences observed for the CHG pattern given that the conclusions focus only on CHH? There appears to be a slight difference not just between

the SAM and leaf but also between the reproductive and vegetative SAMs in Fig 2b, could, is there an explanation for this? Statistically one would expect some variation between all the groups for all the graphs in Fig2 and other graphs in this study. How did the authors draw the line between noise and true variation?

7. Line 105, There also seem to be differences in the methylation of the upstream and downstream surroundings for TEs. in 2b not just at the edges.

8. Line 116, Primarily CHH hypermethylation might be a tool to control TE activity as you state later but could it possibly be multifunctional, since only a portion lines up with TEs?

9. Line 118, Is this an overrepresentation of MITEs compared to their abundances in the entire TE population in the genome in general?

10. Line 125, Authors can just compare the inflorescence and leaf methylation patterns directly and come to a similar conclusion.

11. Line 152, Its pretty nice how well the Fig 4d RPM values and Fig2b avg methylation rate graphs line up with each other. There are tradeoffs with separate vs combined graph although it might be something to consider if you are looking to save space.

12. Line 173, Authors tested overlap of osdrm2 hypoDMRs with both RehyperDMR and LeafhypoDMR but they only seem to mention leaf results despite the former showing a similarly small overlap.

13. Do the authors think there is sufficient data to tie RdDM to being the primary cause of the methylation changes in the SAM, given the somewhat ambivalent evidence in Fig. 4f? while Fig. 3e also seems to point to a more complicated picture that cannot be linked to any one mechanism.

14. Line 190, It could be helpful to have a graph directly combining and comparing the methylation rates of germ cells and the SAM.

15. Line 190, The majority of TEs highly methylated in egg cells do not have hypermethylation in the SAM transition. Maybe there should be more explanation of what sets the hypermethylated egg TE set apart from the rest. Otherwise the overlap with the CHH floral hypermethylated TE set might seem a least partially by chance. In this context, linking these two temporally uncoupled (distant) events based on partial overlaps could be a mis or over interpretation.

16. Line 200, The conclusion might be expanded with a little more biological context/background to complement the observations. Authors must explain why MITEs are especially targeted for methylation based silencing of this mechanism?

17. Findings of the data linking these methylation events to a protective function by silencing TE activity during key events appears to be one of the central findings of the study but is only really brought up at the conclusion. Maybe this model could be introduced earlier/more prominently to allow readers to understand what idea is being reinforced.

Figures

Fig2a. There appears to be a slight difference in the methylation rates for CHG in the most highly methylated genes. Do authors have any theories on this? In some of the graphs there appears to be a central mass of green dots with few overlapping black dots and I'm not sure what the significance of that is or if that is actually a mass of both green and black dots that could be better visualized with a different color scheme.

Fig4b. Are the numbers across the top of the charts refer to replicate datasets?

Fig. 5a. For metagene plots do the authors have any particular reason for using cutoffs of 3/4kb within the gene or is it just arbitrary?

Reviewer #4 (Remarks to the Author):

The study is dealing with DNA methylation in rice shoot apical meristem and proteomic analysis (subject of my review) forms only minor part of the whole study. The authors supported their

conclusions based on DNA and RNA sequencing by proteomic analysis using two argonaut proteins OsAGO4a and OsAGO4b as examples. The applied proteomic approach and using emPAI for assessment of differences in protein levels within different tissues seems to be adequate for this purpose but I am missing detailed description of analytical procedures and data processing. I do not consider referring to previous publications satisfactory. I also miss basic parameters about identified proteins (score, number of peptides etc.). Please see below my detailed comments.

Supplementary material

Material and methods

Proteomic analysis

The description of proteomic analysis is very brief. The all procedure steps including data processing should be described in detail.

- how much input material was used (each contains 10 shoot apical meristems/replicate was mentioned only in reporting summary, how much leave material???)
- add representative picture of SDS-PAGE separation of SAM/leave protein with marked region excised for analysis (or the whole lane was processed?)
- digestion details (enzyme, time, temperature)
- description of LC-MS/MS analysis (I do not expect you used the exactly the same LC-MS/MS system and the same conditions as in ref 37 published in 2009)
- conditions of database search – you really used exactly the same version of Mascot, the same mass accuracy, etc)
- it should be described how the normalized values of emPAI (or translational rates) were calculated (Fig. 4e and Suppl Table 4), the values to all proteins seems to me very low

Presentation of proteomic results

Suppl. Table 2

There are no details about quality of protein identification (score, number of peptides, sequence coverage) it should be added at least at the form as it presented in ref 37 – Suppl. Table 1.

The proteins OsAGO4a and OsAGO4b is not possible to find in the table according their names used in the main text, there just Gene ID

Suppl. Fig. 4

- the calculation of translational rates is not clear to me from the text „The translational rates were calculated by dividing the emPAI value into the relative expression level.“

Responses to Reviewers' comments:

Reviewer #1 (Remarks to the Author):

The manuscript "DNA methylation is reconfigured at the onset of the reproductive phase in the rice shoot apical meristem" by Higo and co-workers describes the genome-wide DNA methylation pattern of the shoot apical meristem (SAM) before and after the transition to the reproductive phase. The SAM contains the stem cells and gives rise to the germline in flowering plants and hence the epigenetic state of this tissue is of special interest. Using whole genome sequencing of bisulfite-converted genomic DNA (WGBS), the authors observe that it is mainly the DNA methylation pattern in the asymmetric CHH context that shows the highest dynamics in the meristem, with higher levels compared to leaf tissue and increasing methylation during the reproductive transition. This is in accordance with previously published transcriptome data in *Arabidopsis thaliana* and the authors also describe high levels of transcripts involved in establishing CHH methylation in the SAM by analysing publically available data in rice. These transcriptomes point towards a role of the RNA directed DNA methylation (RdDM) pathway to be highly active in the SAM. However, when looking into the WGBS of a mutant in this pathway(s), there is only little overlap between RdDM-dependent CHH methylation and regions that show high CHH methylation in the meristem compared to leaf tissue or during the transition to reproduction within the SAM. Higo and co-workers chose to use leaf tissue of a RdDM mutant instead of SAM material, which makes it difficult to conclude which DNA methylation pathways are responsible for the DNA methylation dynamics in the SAM in rice.

Reply: In response to the reviewer's comment, we investigated the contribution of the RdDM pathway to the regions that show high CHH methylation in the SAM. According to the reviewer's suggestion, we tried to collect SAMs from our *osdrm2* knockout plants to conduct WGBS. However, *osdrm2* knockout plants show pleiotropic phenotypes including severe dwarfism, weakness, and sterility as described (Moritoh et al., 2012). Thus, we were unable to collect enough SAMs (more than 150) to obtain methylome data. We then re-examined our WGBS data for the *osdrm2* mutant. We found that the raw sequence data used in our previous

manuscript were not of high quality and that the genome was insufficiently covered for robust analysis. We offer two reasons for this: 1) the methods used for library preparation, and the sequencing platform we used, were not up-to-date (the experiments were conducted in 2011), and 2) the number of PCR cycles required for library amplification were relatively high (18 cycles), thus decreasing the complexity of sequence reads, which in turn decreases coverage. We therefore tried to improve the analysis by replacing our own sequence data with new sequence data that could be obtained from a public database. To this end, we searched publicly available data for WGBS of *osdrm2* mutants, which yielded the *osdrm2* WGBS data of Feng Tan et al. (2016). The number and quality of reads from these published data seemed better than ours. Using these data, we made density plots showing the frequency distribution of the CHH methylation differences among DMRs (new Figure 4e). In this analysis, we compared methylation differences between the *osdrm2* knockout leaf and the WT leaf. The regions analyzed were Rep-hyper CHH DMRs (i.e., CHH DMRs that are hypermethylated in the reproductive SAM), Leaf-hypo CHH DMRs (hypomethylated in leaf), and Leaf-hyper CHH DMRs (hypermethylated in leaf). The results indicated that the majority of these DMRs are hypomethylated in *osdrm2*. From these analyses, we concluded that OsDRM2 is the major contributor to CHH hypermethylation in the SAM during the floral transition. We have added this improved analysis to the manuscript. We appreciate the valuable comment from the reviewer.

Moritoh S, Eun CH, Ono A, Asao H, Okano Y, Yamaguchi K, Shimatani Z, Koizumi A, Terada R. (2012) Targeted disruption of an orthologue of DOMAINS REARRANGED METHYLASE 2, OsDRM2, impairs the growth of rice plants by abnormal DNA methylation. *Plant J.* 71:85-98.

Tan F., Zhou C., Zhou Q., Zhou S., Yang W., Zhao Y., Li G., Zhou D.-X. (2016) Analysis of chromatin regulators reveals specific features of rice DNA methylation pathways. *Plant Physiol.* 171: 2041-2054.

Moreover, the SAM contains several specialized tissues with diverse functions, such as a quiescent center-like cells as well as a proliferating stem cell population. One can expect that these cells contain a different epigenome, according to their function and distinct roles in providing the future germline, which cannot be revealed when looking at a whole SAM methylome. A recent report described the transcriptome and chromatin H3K4me3 and H3K27me3 profiling of specific meristematic cells during the transition to flowering using an established cell type-specific nuclear isolation protocol (You et al, Nature Communications 2017). Such cell type-specific methylome profiling in gametophytes and seeds in rice have been published, revealing clear differences with implications for the work presented here. Yet, CHH-hypermethylated regions in the egg cell seem largely independent of those that gain methylation in the SAM during the transition to reproduction, questioning the functional role of a putative germline in the SAM. Taken together it is difficult to obtain a clear picture from the current work.

Reply: As the reviewer points out, it is difficult to find a relationship between the role of the rice SAM epigenome and the cell type-specific epigenome of sperm and egg cells because our epigenome data came from whole-SAM sampling and thus lacked cell type specificity. Thus, we took two approaches in our responses to this comment. First, we focused more on meristematic development and the vegetative-to-reproductive transition of the SAM in the Abstract and Introduction. The description about germ cells was reduced. Second, we re-analyzed carefully the relationship between the epigenomes of the SAM and the germ cells. To this end, we made the following improvements to our analysis: we added the methylome data from the rice sperm cell to our analysis (Kim et al., 2019), and then we focused on TEs that are highly methylated in both egg cells and sperm cells. These TEs are methylated at the edges in the SAM. In the egg cells and sperm cells, they are methylated both at the edges and in the bodies (new Fig. 5b). In addition, we found that CHH-hypermethylated regions in the egg cell overlapped with those in the SAM. Among the TEs hypermethylated in egg cells, 62.9% undergo CHH hypermethylation in the SAM in the vegetative and/or reproductive phase (new Fig. 5c). From these new observations we propose that these TEs undergo different changes in DNA methylation in different tissue or cell types: hypermethylation at the edges in the SAM, and hypermethylation at the edges and in the bodies in the egg and sperm cells. These modifications are incorporated into the revised manuscript.

Park K, Kim M, Vickers M, Park JS, Hyun Y, Okamoto T, Zilberman D, Fischer RL, Feng X, Choi Y, Scholten S. (2016) DNA demethylation is initiated in the central cells of Arabidopsis and rice. *Proc. Natl. Acad. Sci. U S A.* 113:15138-15143.

Kim MY, Ono A, Scholten S, Kinoshita T, Zilberman D, Okamoto T, Fischer RL. (2019) DNA demethylation by ROS1a in rice vegetative cells promotes methylation in sperm. *Proc. Natl. Acad. Sci. U S A.* 116:9652-9657

Further remarks are listed below:

1) The authors state several times that the germline is not clearly defined in plants and that this is the main reason for the lack of knowledge about DNA methylation reprogramming. Yet, the authors' work does not contribute to solve this problem. Therefore the respective parts should be re-written, i.e. focus on meristematic development without focusing on the germline that cannot be tackled properly with the current results.

Reply: We accept this comment and, as noted above, have rewritten the Abstract and Introduction to focus on meristematic development and the vegetative-to-reproductive transition of the SAM.

2) The term 'reprogramming' is largely avoided in expense of 'reconfiguration', which is not clearly defined either. Therefore the introduction of a novel term does not help. In most cases 'reprogramming' has been used to explain different aspects of DNA methylation dynamics, including qualitative and quantitative differences, but a unambiguous definition of the term is still missing. It would be helpful to give a brief definition how the authors define 'reconfiguration' in the context of DNA methylation.

Reply: The term 'reprogramming' includes two steps of changes in DNA methylation patterns: the first is the genome-wide demethylation of DNA and the second is the subsequent re-

establishment of new DNA methylation patterns. However, when we consider epigenetic reprogramming in plants, this concept may differ from animals because the disappearance of DNA methylation is not so prominent at the genome-wide level. Thus, in plants, the term epigenetic reprogramming may not be applicable to describe genome-wide changes in DNA methylation, but it can be applied to the changes in DNA methylation of specific features in the genome, i.e., transposons and the set of epialleles during germ cell differentiation. In contrast, the term 'reconfiguration' includes changes in DNA methylation patterns without the massive disappearance of DNA methylation from the genome (Kawakatsu et al. 2017). Based on these differences, we have added the following paragraph in the Introduction:

“The ways to change DNA methylation are categorized as reprogramming and reconfiguration. Reprogramming often includes two sequential changes in DNA methylation patterns: first, demethylation of DNA, and second, re-establishment of new DNA methylation patterns. Genome-wide reprogramming of DNA methylation is essential for proper embryonic development in animals^{1,6}. In plants, DNA methylation reprogramming occurs in pollen, and particularly in transposable elements (TEs), imprinting genes, and some epialleles^{7,8}. In addition, plant germ cell differentiation and fertilization results in changes in DNA methylation, and in most cases these changes are also considered as reprogramming. These changes involve hypermethylation in gametes and hypomethylation in their companion cells (central cells in the female and vegetative cells in the male)^{4,5}. In contrast, the term reconfiguration includes genome-wide changes of DNA methylation patterns without the massive disappearance of DNA methylation⁹. CHH methylation is reconfigured throughout embryogenesis and germination, with methylation increasing during seed development and declining during germination^{9–12}.”

References

1. Smith, Z. D., Sindhu, C. & Meissner, A. Molecular features of cellular reprogramming and development. *Nat. Rev. Mol. Cell Biol.* 17, 139–154 (2016).
4. Kawashima, T. & Berger, F. Epigenetic reprogramming in plant sexual reproduction. *Nature Reviews Genetics* vol. 15 613–624 (2014).
5. Gehring, M. Epigenetic dynamics during flowering plant reproduction: evidence for reprogramming? *New Phytol.* 224, 91–96 (2019).

6. Lee, D. S. et al. An epigenomic roadmap to induced pluripotency reveals DNA methylation as a reprogramming modulator. *Nat. Commun.* 5, (2014).
7. Calarco, J. P. et al. Reprogramming of DNA methylation in pollen guides epigenetic inheritance via small RNA. *Cell* 151, 194–205 (2012).
8. Slotkin, R. K. et al. Epigenetic Reprogramming and Small RNA Silencing of Transposable Elements in Pollen. *Cell* 136, 461–472 (2009).
9. Kawakatsu, T., Nery, J. R., Castanon, R. & Ecker, J. R. Dynamic DNA methylation reconfiguration during seed development and germination. *Genome Biol.* 18, 171 (2017).
10. Bouyer, D. et al. DNA methylation dynamics during early plant life. *Genome Biol.* 18, 179 (2017).
11. Lin, J. Y. et al. Similarity between soybean and Arabidopsis seed methylomes and loss of non-CG methylation does not affect seed development. *Proc. Natl. Acad. Sci. U. S. A.* 114, E9730–E9739 (2017).
12. Narsai, R. et al. Extensive transcriptomic and epigenomic remodelling occurs during Arabidopsis thaliana germination. *Genome Biol.* 18, 172 (2017).

3) The Kernel density plot shows both, gain and loss of CHH methylation when comparing leaf and vegetative SAM tissues (Figure 1b). However, throughout the manuscript only CHH hypermethylation in the SAM is investigated. What about the regions showing CHH hypomethylation in this comparison, especially given that it is very prominent? Also, the meta-gene and meta-TE methylation profiles might be more informative for those TEs containing DMRs (hypo- and hyper-CHH DMRs) in addition to presenting all gene and TE annotations.

Reply: In response to this comment, we analyzed the regions showing CHH hypomethylation in the SAM. These regions are, in other words, hypermethylated in the leaf compared to the SAM, and we thus refer to them as leaf-hyper DMRs. Intergenic TEs comprised 70% of leaf-hyper DMRs. (new Fig. 3b); of these, 41.6 % were MITEs (new Fig. 3c). Meta-TE plots indicated that methylation occurs at the edges of TEs which overlap with leaf-hyper DMRs (Supplementary fig. 4c, right plot). These characteristics are similar to those of TEs hypermethylated in the SAM. These observations have been added to the manuscript. In addition, according to the suggestion from the reviewer, we analyzed DNA methylation for TEs containing DMRs by

metaplots. Briefly, TE borders are highly methylated in TEs containing hyper-CHH DMRs in the reproductive SAM or hyper-CHH DMRs in the leaf. The results are shown in new Supplementary fig.4c.

4) There have been four publications describing the DNA methylation dynamics during embryogenesis, yet only one is cited here. Please also cite the other three articles (Bouyer et al, Genome Biol 2017; Lin et al, PNAS 2017; Narsai et al, Genome Biol 2017).

Reply: We agree with the reviewer, and have cited these additional references in the text.

5) Controls are missing that monitor the transition from vegetative to reproductive state in the SAM. Marker genes specific for each phase should be tested in RT-PCR experiments on the material used for WGBS.

Reply: In response to this comment, we measured the expression of a known reproductive SAM-specific gene (*OsMADS15*) in the SAMs at the same stage as those used in WGBS. The result indicated clear induction of *OsMADS15* in the reproductive SAM, and is shown in Supplementary fig. 1b.

6) In the analysis of DMRs affecting TEs I was missing a genome-wide profile of the different TE families/classes (Figure 3d). In the current form, one cannot conclude if a certain TE family, such as MITEs, are enriched, or if they mainly reflect the global average of TEs as present in the genome.

Reply: We performed additional analyses to determine the genome-wide profiles of different TE families in response to this comment. MITEs occupy 7% of the whole genome in length, but they occupy 41% of the total length of DMRs that are hypermethylated in the reproductive SAM relative to the vegetative SAM. This new result supported our previous conclusion and is shown in lines 158 – 161 and new Table 1. "Of the TE-overlapping Rep-hyperDMRs, 41.1% corresponded to miniature inverted-repeat transposable element (MITE)-type TEs (Fig. 3c) in nucleotide length, although MITEs comprise only 7.0% of the total length of TEs in the rice genome (Table 1)."

7) The authors state that the MITEs are located close to genes, reflecting the DNA methylation pattern seen for genes (line 120-122). This could and should be shown as a graph in the figure.

Reply: In response to this comment, we analyzed the pattern of DNA methylation for genes with or without insertions of MITEs. Metagene plots showed that genes with MITE insertions displayed higher CHH methylation than genes without MITE insertions. The new results are shown in Supplementary fig. 3.

8) It is surprising that leaf vs. vegetative SAM DMRs do not overlap with vegetative vs. reproductive SAM DMRs (Figure 3e). It would be interesting to see the global DNA methylation pattern, i.e. to which extent does the dynamic of CHH methylation affect TEs globally or only specifically? Also, why do the authors analyse the vegetative vs. reproductive SAM DMRs with respect to their association with TEs, but not the leaf vs. vegetative SAM CHH DMRs? This should be completed.

Reply: We think the reason why leaf vs. vegetative SAM DMRs do not overlap with vegetative vs. reproductive SAM DMRs is as follows: RdDM pathway have different methylation preferences in different developmental contexts, and so the RdDM pathway methylates different sets of TEs in leaf, vegetative SAM and reproductive SAM. According to the reviewer's suggestion, we analyzed whether CHH methylation affects TEs globally or locally. We made density plots of the differences in CHH methylation between vegetative SAM vs leaf and vegetative SAM vs reproductive SAM for regions overlapping with MITEs (Supplementary figure 5). These plots revealed a slight shift of the central peaks: hypermethylation of the vegetative SAM in the vegetative SAM vs leaf comparison, and hypermethylation of the reproductive SAM in the vegetative SAM vs reproductive SAM comparison. The peak shifts in the density plots are interpreted as reflecting global changes in methylation differences, suggesting that CHH methylation affects TEs globally in different tissues. The new results have been added as Supplementary figure 5 and are explained on lines 207–215 in the new manuscript. We also analyzed the CHH DMRs between leaf and vegetative SAM with respect to their association with TEs, and found that the majority of the leaf CHH hypo DMRs also

overlapped with TEs, especially MITEs, as shown in the analysis of reproductive SAM hyper CHH DMRs. We have added the results to Figure 3b and 3c.

9) The authors state that RdDM methylates CHH in TE bodies (line 135-136). This is misleading as the majority of RdDM-dependent CHH methylation takes place at TE borders and the TE bodies are mainly CHH methylated by CMT2 (Stroud et al, Nature Struc. Mol Biol 2014). However, small TEs are often mainly targeted by RdDM, thus covering also the TE body as a consequence. This needs to be clarified in the manuscript.

Reply: We accept this comment and have corrected the main text (lines 217–219 in the new manuscript) as follows: " The RdDM pathways are well-known mechanisms to methylate CHH sites at the edges of TEs, and CMT2 methylates CHH sites in the bodies of TEs¹⁷ (Fig. 4a).."

10) Given that DRM2-dependent CHH methylated regions and leaf vs. vegetative SAM- as well as vegetative vs. reproductive SAM-DMRs do not overlap much (Figure 4f), the authors should also analyse *cmt2* mutants. Moreover, instead of using leaves, the SAM should be used as material.

Reply: Our response to this comment is the same as that to a major comment above. We reiterate it here for convenience. Because we could obtain new results that showed overlap between DRM2-dependent CHH-methylated regions and SAM-DMRs, we did not analyze *cmt2* in this study.

Reply: In response to the reviewer's comment, we investigated the contribution of the RdDM pathway to the regions that show high CHH methylation in the SAM. According to the reviewer's suggestion, we tried to collect SAMs from our *osdrm2* knockout plants to conduct WGBS. However, *osdrm2* knockout plants show pleiotropic phenotypes including severe dwarfism, weakness, and sterility as described (Moritoh et al., 2012). Thus, we were unable to collect enough SAMs (more than 150) to obtain methylome data. We then re-examined our WGBS data for the *osdrm2* mutant. We found that the raw sequence data used in our previous manuscript were not of high quality and that the genome was insufficiently covered for robust analysis. We offer two reasons for this: 1) the methods used for library preparation, and the

sequencing platform we used, were not up-to-date (the experiments were conducted in 2011), and 2) the number of PCR cycles required for library amplification were relatively high (18 cycles), thus decreasing the complexity of sequence reads, which in turn decreases coverage. We therefore tried to improve the analysis by replacing our own sequence data with new sequence data that could be obtained from a public database. To this end, we searched publicly available data for WGBS of *osdrm2* mutants, which yielded the *osdrm2* WGBS data of Feng Tan et al. (2016). The number and quality of reads from these published data seemed better than ours. Using these data, we made density plots showing the frequency distribution of the CHH methylation differences among DMRs (new Figure 4e). In this analysis, we compared methylation differences between the *osdrm2* knockout leaf and the WT leaf. The regions analyzed were Rep-hyper CHH DMRs (i.e., CHH DMRs that are hypermethylated in the reproductive SAM), Leaf-hypo CHH DMRs (hypomethylated in leaf), and Leaf-hyper CHH DMRs (hypermethylated in leaf). The results indicated that the majority of these DMRs are hypomethylated in *osdrm2*. From these analyses, we concluded that OsDRM2 is the major contributor to CHH hypermethylation in the SAM during the floral transition. We have added this improved analysis to the manuscript. We appreciate the valuable comment from the reviewer.

Moritoh S, Eun CH, Ono A, Asao H, Okano Y, Yamaguchi K, Shimatani Z, Koizumi A, Terada R. (2012) Targeted disruption of an orthologue of DOMAINS REARRANGED METHYLASE 2, OsDRM2, impairs the growth of rice plants by abnormal DNA methylation. *Plant J.* 71:85-98.

Tan F., Zhou C., Zhou Q., Zhou S., Yang W., Zhao Y., Li G., Zhou D.-X. (2016) Analysis of chromatin regulators reveals specific features of rice DNA methylation pathways. *Plant Physiol.* 171: 2041-2054.

11) The strong contribution of sRNAs in the leaf with >25nt size is suggested to result from PolIV primary transcripts (line 142-144 and Figure 4c). However, transcript levels of PolIV subunits are low in leaf tissue, but very high in the SAM, questioning the given explanation for this observation.

Reply: We accept this comment and, since the evidence for PolIV primary transcripts is lacking, have deleted the sentence from the text. We analyzed our smRNA-seq data, but we could not clearly show that the leaf-expressed >25-nt smRNAs represent the precursors of SAM-expressed 24-nt smRNAs.

12) In line 155-158 the authors cite a publication, suggesting mobility of 24nt sRNAs from surrounding tissue to reinforce TE silencing, which was shown for 21-22nt sRNAs in pollen, but not for sRNAs in the ovule or seed as mentioned in the manuscript. This needs to be corrected.

Reply: We agree with this comment and have changed a sentence to correct the differences of sRNA length and the identity of the tissues. We made the following modifications on lines 364–366 of the new manuscript: "In Arabidopsis, DNA methylation in the sperm cell is regulated by mobile 21–22-nt smRNAs from its companion cell, the vegetative cell^{7,8,29}."

13) The sRNA profile between vegetative and reproductive SAM does not change much, which is in agreement with the stable expression patterns of the RdDM components. How does the sRNA distribution look like for those sRNAs that target DMRs detected in the SAM during phase transition?

Reply: To answer the reviewer's question, we analyzed sRNA distribution for TEs with CHH-hypermethylated DMRs in the reproductive SAM and found a slight increase in sRNA accumulation at the edges of these TEs. We have added the results to Supplementary Fig. 6. We add following text to lines 241 – 248 in the new manuscript:

"These results suggest that the increase in 24-nt smRNAs contributes to the increase in CHH methylation at the edge of TEs during the vegetative-to-reproductive transition in the SAM (Fig. 2b and c). To assess this more closely we analyzed smRNA profiles along the TEs overlapping

with Rep-hyper DMRs, which revealed that smRNA increased in the reproductive SAM at the edges of these TEs (Supplementary Fig. 6). Together, these results imply that 24-nt smRNAs in the SAM contribute to CHH hypermethylation in the SAM.”

14) The authors mention a SAM proteome, but the only proteins shown are OsAGO4a and OsAGO4b. Therefore, it is unclear how these two proteins behave in comparison with the overall pattern. For instance, it was suggested that translational control might be involved in the transition from vegetative to reproductive development in the SAM (You et al, Nature comm. 2017). The authors need to provide further information on this aspect.

Reply: We analyzed our proteome data for the SAM and found that there is no global increase in protein levels in the rice SAM. This result suggested that protein level of OsAGO4b is positively regulated in the reproductive SAM through an unknown mechanism. We made a scatter plot to show the overall pattern of changes in the proteome in Supplementary Fig. 7. Through this analysis we found that normalization should be refined: in the refined normalization, AGO4b is increased in the vegetative-to-reproductive transition of the SAM and this is reflected in Fig. 4d.

15) In line 178-180 the authors propose another de novo methylation pathway beside RdDM to contribute to elevated CHH methylation levels in the SAM. It should be taken into account that there are non-canonical RdDM pathways that have been studied in more detail in Arabidopsis, but all rely on sRNAs (see Cuerda-Gil & Slotkin, Nature Plants 2016 for example).

Reply: We agree with this comment and have changed the text to explain the possible contribution from non-canonical RdDM pathways, with the suggested citation, on lines 278–280 of the new manuscript.

16) The DNA methylation profile shows higher levels in the egg cell compared to all other cells or tissues investigated so far, including the SAM. This, together with the observation that CHH hypermethylation in this cell does not correspond to DNA methylation dynamics in the SAM, challenges the reprogramming concept and the order of events. Moreover, it questions the approach to use the entire SAM in order to reveal specific DNA methylome reprogramming. The authors state that the CHH methylation profile is qualitatively different in egg cells and in the SAM, yet the sRNA profile in the SAM matches the CHH methylation profile in the egg cell. Therefore it is possible that certain cells within the SAM show a similar CHH methylation profile compared to the egg cells, and others maintain CHH methylation mainly at the TE borders.

Reply: We thank the reviewer for these noteworthy and insightful remarks on the overlap of the SAM methylome with the egg cell methylome. As the reviewer suggests, certain cells within the SAM may show a similar CHH methylation profile to the egg cells, and this possibility has now been included in the text. Our response to this comment is partly overlapping with the response to the comment 1 from Reviewer 3. Our responses are summarized as follows and have been added to the text (lines 344–357): "Certain cells within the SAM may show a similar CHH methylation profile to the egg cells. They may represent the precursors of the germ line in the SAM. Germ cells develop from L2 of the SAM in *Arabidopsis*. Thus, it is possible that L2 cells express RdDM pathway genes to confer a CHH methylation pattern similar to that in the germ cells. If this is the case, L2 cells should express different levels of RdDM genes. To examine this, we analyzed the published layer-specific transcriptome data for *Arabidopsis* SAM cells³⁴ (Supplementary Fig. 8). This indicated that RdDM pathway_genes showed no specific enrichment in any cell type, including L2. Thus, from the expression patterns of RdDM genes in L2, we could not conclude that L2 cell express a specific pattern of RdDM genes to establish a germ cell-specific methylation profile of the genome. In addition, unlike the *Arabidopsis* SAM, L2 cannot be clearly defined by cell lineage analysis in the SAM of grass species including rice³⁵. Further investigation will be required to reveal the cellular identity and contribution of subepidermal layer(s) of the SAM in grass species."

In addition, to evaluate the contribution of the CHH hypermethylation in the SAM to the methylome of the germ cells, we analyzed the methylome of the egg cells and sperm cells (new Figure 5; analysis in the previous manuscript was restricted to the egg cell methylome).

Examination of TEs highly methylated in both egg cell and sperm cell showed that 62.9% of the TEs highly methylated in egg or sperm cells overlap with the TEs hypermethylated in the SAM; of these, 33% are highly methylated in the vegetative SAM and 28.6% are further methylated in the reproductive SAM. The methylation takes place at the borders and in the bodies of TEs in the egg cell and sperm cell, but only at the borders in the SAM, suggesting that these TEs are affected by two rounds of changes in genome-wide DNA methylation, namely the TE border in the SAM and both the border and the body in the egg cell and sperm cell.

Reviewer #2 (Remarks to the Author):

Higo et al show that the reproductive shoot apical meristem (SAM) of rice is genome-wide hypermethylated for the CHH context in comparison to the vegetative SAM.

Furthermore, their results indicate that the methylation patterns are modified in some genomic regions between vegetative and reproductive SAM. Therefore, this process is not only a hypermethylation of the genome, but also a structural change. So far, genome-wide DNA methylation is described for female and male germ cell formation and for developing seeds, but our knowledge remained elusive for the progenitor cells of germ cells in regard to changing DNA methylation.

The authors also provide a deeper insight for the mentioned elevated CHH hypermethylation by showing that both transposable elements (TE) and protein-coding genes are affected. They discover that MITE transposons are more often hypermethylated than other TEs in reproductive SAM and they suggest that the genome is protected by this mechanism. This suggest that the MITEs may be the major threat during sexual reproduction in rice, possibly because MITEs are often in vicinity to protein-coding genes. The role of MITEs should be analysed in more detail in the manuscript, by analyzing DNA methylation mutants and activation of MITEs; and/or in another rice accession with higher or lower amounts of MITEs and DNA methylation analyses.

Reply: The reviewer's comments are related to the role of DNA methylation of MITEs, especially in the SAM. As suggested by the reviewer, analyzing DNA methylation mutants and activation of MITEs is a possible way to examine this. However, DNA methylation mutants in

rice, such as *osdrm2*, show severe dwarfism and weakness, and thus we were unable to collect enough SAMs for transcriptome and DNA methylome analysis. The reviewer notes that another way to analyze this is to examine transcriptome and DNA methylome in the SAM of another rice accession with higher or lower amounts of MITEs. An increase in the number of MITEs in specific accessions will have occurred during the relatively short duration of their evolution, and at present the MITE bursts are suppressed in these accessions. We thus could not analyse the relationship between DNA methylation and active MITE mobilization. Considering these situations, we analyzed the effect of the presence or absence of MITEs on DNA methylation levels by comparing the CHH methylation profiles of genes with or without MITE insertions. Metagene plots showed that the CHH methylation on MITEs located around genes reflects the CHH methylation pattern for these genes: higher methylation in the vicinity of genes with MITE insertions, and *vice versa*. This indicates that MITE insertions reflect higher CHH methylation in the vicinity of the genes. The new results are shown in Supplementary Fig. 3.

The authors show that egg cells have an even higher level of CHG and CHH DNA methylation than the reproductive SAM, which will give raise to the egg cells during the plant's life cycle. That indicates that the change between vegetative and reproductive SAM is characterized by a change mostly in CHH methylation while the maturation of egg cells involve an even higher DNA methylation in the CHH and the CHG context. However, the biological reason for those changes remains elusive. Why do the progenitor cells of the germ cells (in the SAM) are not protected from TE activity during all stages of development? This should be discussed.

Reply: As the reviewer indicates, the egg cell showed higher DNA methylation than the progenitor cell in the SAM. We also confirmed that DNA methylation was higher in rice sperm cell than in the SAM (new Figure 5); thus, germ cells methylate DNA to a higher degree than SAM. Because we have shown previously that TE expression is reduced by the reproductive transition in the rice SAM (Tamaki et al., 2015), the level of TE methylation in the reproductive SAM may be sufficient to silence TE activation at this stage. In contrast, we speculate that the germ cells require stricter regulation of TEs, and this requirement is reflected by the higher levels of methylation in the CHH and CHG contexts. The above considerations are summarized as follows and inserted into the Discussion (lines 339–342): "We have shown that TE

expression is reduced by the reproductive transition of the rice SAM²; hypermethylation may contribute to this silencing. Further hypermethylation in the CHH and CHG contexts in the germ cells may reflect a strict requirement for TE silencing in these cells."

The activity of the RNA dependent DNA methylation (RdDM) can be partially responsible for the hypermethylation between vegetative and reproductive SAM according to the authors. They performed smRNA-seq and proteome analysis. The results of Higo et al. show that 24nt smRNA are slightly enriched in reproductive SAM in comparison to the vegetative SAM. However, Figure 4d shows that 24nt smRNA is less abundant in the TE bodies in reproductive SAM compared to vegetative SAM. This should be discussed.

Reply: We thank the reviewer for this insightful remark on the results of the small RNA analysis. As the reviewer suggests, 24-nt smRNAs are less abundant in the TE bodies in the reproductive SAM. The mechanism for this is not known at present, but we speculate that the transcription of small RNA precursors by RNA polymerase IV and/or their processing by RDR2 and DCL3 are slightly attenuated for the region corresponding to the long TE bodies. Because the total fraction of 24-nt smRNA is higher in the reproductive SAM (Figure 4c), this reduction in TE bodies is compensated for by the increase at the TE edges. This is summarized in the following addition to the main text (lines 238–241): " In TE bodies, however, 24-nt smRNAs are less abundant in the reproductive SAM than in the vegetative SAM (Fig. 2c), perhaps because the transcription of small RNA precursors by RNA polymerase IV or their processing by RDR2 and DCL3 is slightly attenuated in regions corresponding to these long TE bodies."

Furthermore, the authors discover that AGO4 is higher expressed in the reproductive SAM. However, the methylation sites catalyzed by the AGO4 associated DRM2 DNA de novo methyltransferase are just partially overlapping with the hypermethylated sites in the reproductive SAM. The authors suggest that DDM1 which is higher expressed in the SAM tissue also plays a role for the increased hypermethylation.

In the end, the authors could not completely explain the mechanism of the hypermethylation. In my point of view, this should be addressed/discussed further.

Reply: In response to the reviewer's comment, we investigated the contribution of the RdDM pathway to the regions that show high CHH methylation in the SAM. we tried to collect SAMs from our *osdrm2* knockout plants to conduct WGBS. However, *osdrm2* knockout plants show pleiotropic phenotypes including severe dwarfism, weakness, and sterility as described (Moritoh et al., 2012). Thus, we were unable to collect enough SAMs (more than 150) to obtain methylome data. We then re-examined our WGBS data for the *osdrm2* mutant. We found that the raw sequence data used in our previous manuscript were not of high quality and that the genome was insufficiently covered for robust analysis. We offer two reasons for this: 1) the methods used for library preparation, and the sequencing platform we used, were not up-to-date (the experiments were conducted in 2011), and 2) the number of PCR cycles required for library amplification were relatively high (18 cycles), thus decreasing the complexity of sequence reads, which in turn decreases coverage. We therefore tried to improve the analysis by replacing our own sequence data with new sequence data that could be obtained from a public database. To this end, we searched publicly available data for WGBS of *osdrm2* mutants, which yielded the *osdrm2* WGBS data of Feng Tan et al. (2016). The number and quality of reads from these published data seemed better than ours. Using these data, we made density plots showing the frequency distribution of the CHH methylation differences among DMRs (new Figure 4e). In this analysis, we compared methylation differences between the *osdrm2* knockout leaf and the WT leaf. The regions analyzed were Rep-hyper CHH DMRs (i.e., CHH DMRs that are hypermethylated in the reproductive SAM), Leaf-hypo CHH DMRs (hypomethylated in leaf), and Leaf-hyper CHH DMRs (hypermethylated in leaf). The results indicated that the majority of these DMRs are hypomethylated in *osdrm2*. From these analyses, we concluded that OsDRM2 is the major contributor to CHH hypermethylation in the SAM during the floral transition. We have added this improved analysis to the manuscript. We appreciate the valuable comment from the reviewer.

Moritoh S, Eun CH, Ono A, Asao H, Okano Y, Yamaguchi K, Shimatani Z, Koizumi A, Terada R. (2012) Targeted disruption of an orthologue of DOMAINS REARRANGED METHYLASE 2, OsDRM2, impairs the growth of rice plants by abnormal DNA methylation. *Plant J.* 71:85-98.

Tan F., Zhou C., Zhou Q., Zhou S., Yang W., Zhao Y., Li G., Zhou D.-X. (2016) Analysis of chromatin regulators reveals specific features of rice DNA methylation pathways. *Plant Physiol.* 171: 2041-2054.

Additional points:

Line 123: This hypothesis should be further discussed: Are there other TEs that are close to genes? Do these have the same distribution of CHH methylation than MITEs? Are there similar observations known in Arabidopsis or even animals?

Reply: It has been reported that several families of TEs are located close to genes in rice genome (e.g., Li et al., 2017, Dubin et al., 2018), and this has been added to the text (lines 163 - 165 in the revised manuscript). We analyzed the distribution of CHH methylation of TEs other than MITEs, and found that the distributions of CHH methylation in these TEs were similar to that in MITEs. For short TEs, they are methylated in the body regions; for longer TEs, they are methylated at the edges. The differences of methylation levels in different tissues showed the same pattern as MITEs did: low in the leaf, high in the vegetative SAM, and higher in the reproductive SAM. We have presented these data as Supplementary Figure 2. It has also been reported that the frequency distribution of TEs is high around 100–300 bp from genes in *Arabidopsis* (Ahmed et al., 2011, Quadrana et al., 2016). A similar tendency for TEs to insert near genes is observed in the human genome (Lowe et al., 2007). These observations have been added to the manuscript (lines 325 - 326) as follows: “A tendency to insert near genes is also observed for *Arabidopsis* and human TEs¹⁹⁻²¹”

References

X. Li, K. Guo, X. Zhu, P. Chen, Y. Li, G. Xie, L. Wang, Y. Wang, S. Persson, L. Peng (2017) Domestication of rice has reduced the occurrence of transposable elements within gene coding regions. *BMC Genomics* 18: 55

Dubin MJ, Mittelsten Scheid O, Becker C (2018) Transposons: a blessing curse. *Curr Opin Plant Biol.*42:23-29.

Ahmed I, Sarazin A, Bowler C, Colot V, Quesneville H (2011) Genome-wide evidence for local DNA methylation spreading from small RNA-targeted sequences in Arabidopsis. *Nucleic Acids Res.* 39:6919–6931,

Quadrana L, Silveira AB, Mayhew GF, LeBlanc C, Martienssen RA, Jeddloh JA, Colot V (2016) The Arabidopsis thaliana mobilome and its impact at the species level. *eLife* 2016;5:e15716

Lowe, C. B., Bejerano, G. and Haussler, D. (2007). Thousands of human mobile element fragments undergo strong purifying selection near developmental genes. *Proc. Natl. Acad. Sci. USA* 104, 8005-8010. doi:10.1073/pnas.0611223104

Line 124: The reason why Rep-hyperDMRs and Leaf-hypoDMR are compared should be explained more: figure 1a shows high CHH methylation areas of Rep but the hypermethylated areas of leaf do not overlap very often.

Reply : In this analysis, we tried to examine the relationship between two regions targeted by DNA methylation activity, one for regions methylated during the vegetative-to-reproductive transition in the SAM and the other for regions methylated during leaf differentiation from the SAM. In the comparison between the vegetative SAM and reproductive SAM, we found that the majority of DMRs are Rep-hyper DMRs, suggesting that a mechanism responsible for this process functions to methylate specific regions in the reproductive SAM more strongly than in the vegetative SAM. In contrast, the comparison between the vegetative SAM and leaf indicated that the majority of DMRs are leaf-hypo DMRs, suggesting that a mechanism responsible for this process functions to methylate specific regions in the vegetative SAM to a higher level than in the differentiated leaf. We tried to examine whether these two processes share a similar mechanism by comparing the overlap between the regions to be targeted for DNA methylation. These comments are summarized in the following addition to the main text (lines 182–192): “Our results suggest that CHH methylation generally declines during leaf differentiation from the vegetative SAM, but rises upon the vegetative-to-reproductive transition. To examine whether these two processes share DMRs, we analyzed the overlap between Leaf-hypoDMRs and Rep-hyperDMRs. Leaf-hypoDMRs represent regions hypomethylated in the leaf relative to

the vegetative SAM, and thus reflect regions whose CHH methylation declines during leaf differentiation from the vegetative SAM. Rep-hyperDMRs represent regions hypermethylated in the reproductive SAM relative to the vegetative SAM, and thus reflect regions whose CHH methylation rises upon the vegetative-to-reproductive transition of the SAM. Only a part of these DMRs overlapped (Fig. 3d), suggesting that CHH methylation varies among different developmental processes.”

Line 166: It is unclear what is referred to: certain cell types of the SAM? Please explain.

Reply: We have conducted our proteome analysis using whole SAM tissue, i.e., the same tissue as for the DNA methylome, transcriptome, and small RNA transcriptome analyses we have conducted in this study.

Line 178: I think it would be better to explain why leaf-hypoDMRs and *osdrm2*-hypoDMR are compared. The *de novo* DNA methyltransferase DRM2 is unlikely the reason for the hypomethylation in leaf, but it may be for the hypermethylation in Reproductive SAM.

Reply : In this analysis we tried to examine whether DRM2 contributes to the hypermethylation of genomic DNA in the vegetative SAM more so than in the leaf. Leaf-hypo DMRs indicates regions hypomethylated in the leaf; in other words, regions hypermethylated in the SAM. Thus, Leaf-hypo DMRs is the same as SAM-V-hyper DMRs compared to leaf. On the other hand, *osdrm2*-hypo DMRs indicates regions hypomethylated in *osdrm2* mutants; these regions are methylated by OsDRM2 enzymatic activity in wild-type. Overlap between Leaf-hypo DMRs and *osdrm2*-DMRs means the overlap between the regions hypermethylated in the SAM and regions methylated by OsDRM2, representing regions where OsDRM2 methylates in the SAM more strongly than in the leaf. We refined our results using reported methylome data for *osdrm2* plants, and found that almost all of the SAM Rep-hyper CHH DMRs and Leaf-hypoDMRs are more extensively hypomethylated in *osdrm2* than in WT (new Fig. 4e). These comments are summarized in the following addition to the main text (lines 267–272): "we examined whether OsDRM2 contributes to hypermethylation in the SAM compared with the leaf by examining the overlap between Rep-hyper DMRs (regions hypermethylated in the reproductive SAM

compared to the vegetative SAM), leaf-hypo DMRs (regions hypermethylated in the SAM compared to the leaf), leaf-hyper DMRs (regions hypermethylated in the leaf compared to the vegetative SAM) and *osdrm2*-DMRs (regions methylated by OsDRM2).”

Suppl.page3 Line 18: Where are those methods described?

Reply : The response to this comment is the same as that to a major comment raised by reviewer 4. We reiterate it here for convenience.

- how much input material was used (each contains 10 shoot apical meristems/replicate was mentioned only in reporting summary, how much leave material???)

We have added following description to Supplementary methods: “We collected 10 shoot apical meristems from each of 10 different plants for each replicate of the analysis. We also sampled and mixed three leaves from each of three different plants for each replicate.”

- add representative picture of SDS-PAGE separation of SAM/leave protein with marked region excised for analysis (or the whole lane was processed?)

We added SDS-PAGE of the SAM and leaf protein as Supplementary Figure 9. Whole lane was processed.

- digestion details (enzyme, time, temperature)

We have added the following descriptions to Supplementary methods.

“We extracted proteins from the samples with Laemmli sample buffer and incubated them at 65°C for 15 min, and then subjected them to SDS–PAGE [acrylamide concentration 10.5% (w/v)] (Supplementary Fig. 9). After staining by Flamingo (BioRad, Hercules, CA, USA), we sliced each lane of the gel into four parts of equal length. We washed the sliced pieces twice with HPLC-grade water with 30% (v/v) acetonitrile (Kanto Chemical, Tokyo, Japan), once with 100% acetonitrile, and then dried them in a vacuum concentrator. We treated the dried gel

pieces with 2 μl of 0.5 $\mu\text{g } \mu\text{l}^{-1}$ trypsin (sequence grade; Promega, Madison, WI, USA) in 50 mM ammonium bicarbonate⁴⁴, followed by incubation at 37°C for 16 h. We recovered the digested peptides from the gel pieces by extracting twice with 20 μl of 5% (v/v) formic acid/50% (v/v) acetonitrile. The peptides were combined and dried in a vacuum concentrator.”

- description of LC-MS/MS analysis (I do not expect you used the exactly the same LC-MS/MS system and the same conditions as in ref 37 published in 2009)

We have added the following description to Supplementary methods.

“We performed LC-MS/MS analyses with an LTQ-Orbitrap XL-HTC-PAL-Paradigm MS4 system. Peptides were digested with trypsin and then loaded on the column (75 μm internal diameter, 15 cm; L-Column, CERI, Auburn, CA, USA). At this step we used a Paradigm MS4 HPLC pump (Michrom BioResources) and an HTC-PAL autosampler (CTC analytics, Zwingen, Switzerland). We prepared two buffers, 2% (v/v) acetonitrile and 0.1% (v/v) acetic acid (buffer A) and 90% (v/v) acetonitrile and 0.1% (v/v) acetic acid (buffer B). We applied a linear gradient from 5 to 45% by mixing buffers A and B, for 25 min, and after elution from the column we introduced the eluted peptides into an LTQ-Orbitrap mass spectrometer (Thermo Fisher Scientific, Bremen, Germany). The flow rate was 300 nl min^{-1} and the spray voltage 2.0 kV. We set the range of MS scanning as m/z 200–2,000, and the top three peaks were subjected to MS/MS analysis.”

- conditions of database search – you really used exactly the same version of Mascot, the same mass accuracy, etc)

We have added the following description to Supplementary methods.

“We compared the spectra with the protein database OSMSU (<http://rice.plantbiology.msu.edu>) using the MASCOT server (version 2.4.1, Matrix Science, London, UK). We used the following parameters for the MASCOT search: set off the threshold at 0.05 in the ion score cut-off, peptide tolerance at 10 p.p.m., MS/MS tolerance at ± 0.5 Da, peptide charge of 2 + or 3 +,

trypsin as enzyme allowing up to one missed cleavage, carbamidomethylation on cysteines as a fixed modification and oxidation on methionine as a variable modification.”

-it should be described how the normalized values of emPAI (or translational rates) were calculated (Fig. 4e and Suppl Table 4), the values to all proteins seems to me very low

We have added the following descriptions to Supplementary methods.

“We calculated the “normalized emPAI value” in Figure 4d by dividing emPAI of each protein by the sum of emPAI values of all the proteins in each sample^{45,46}. This normalization was applied to compare between different samples with different sums of emPAI values.”

Reviewer #3 (Remarks to the Author):

A dedicated germ line that is established early during embryonic development produces gametes in animal development. However, in plant development, the gamete forming cells are thought to arise from the L2 layer of SAMs. Authors analyze CHH methylation status in both vegetative and reproductive SAMs, which was found to be higher. Authors combine transcriptome and proteome data sets to show that the higher CHH methylation requires the activity of RNA-dependent DNA methylation pathway. They further found that DNA methylation patterns at transposons in the SAM partially overlapped with methylation patterns observed in reprogramming egg cells. Based on these observations, authors argue that reconfiguration of DNA methylation precedes germ cell differentiation in SAMs. Authors data showing higher levels of CHH methylation very early in vegetative SAMs in showing that reconfiguration of the methylation occurs very early during development is interesting. The following points must be clarified before linking observed methylation changes in SAMs to germ cells.

Comments

1. It is well documented that germ cells are specified from L2-layer derived cells in SAMs. Since authors use the whole SAMs for their analysis, it is not clear whether the observed methylation patterns are layer specific or occurs in all cell layers. What if the observed methylation changes occurred in all cell layers except in the L2 layer? I understand that it may be technically difficult to isolate cells from different layers in rice SAMs. Therefore, authors must be careful in interpreting these results and bring this point into their discussion. The layer specific transcriptome for Arabidopsis SAM cells is available (Yadav et al., Development 2014, 141(13):2735-44; Yadav et al., PNAS 2009, 106(12):4941-6). Authors must examine these data sets to see whether the RNA-dependent DNA methylation pathway components are enriched in any specific cell layers. I understand that the regulatory RNAs can move across cells, which can result in non-cell autonomous effects, but efforts should be made to see whether the pathways regulating methylation are layer-specific.

Reply: In response to the reviewer's comment, we examined the expression patterns of genes that encode enzymes responsible for the RdDM pathway in specific layers of the *Arabidopsis* SAM, and found that RdDM genes are expressed at similar levels across different cell layers, including the L2 layer (Yadav et al., Development 2014, 141(13):2735-44). We have presented these data as Supplementary Figure 8. The L2 layer of the SAM in grass species is different from that of *Arabidopsis*, in which the L2 layer can be clearly defined and anticlinal division takes place in the cells. In contrast, in grass species, cell lineage analysis has indicated that the SAM is organized into an outermost L1 and inner cell layers; periclinal divisions occur in the cells of the second layer from L1. This difference in the SAM anatomy is also discussed. These points are summarized as follows and have been added to the text (lines 346–359):

“Certain cells within the SAM may show a similar CHH methylation profile to the egg cells. They may represent the precursors of the germ line in the SAM. Germ cells develop from L2 of the SAM in *Arabidopsis*. Thus, it is possible that L2 cells express RdDM pathway genes to confer a CHH methylation pattern similar to that in the germ cells. If this is the case, L2 cells should express different levels of RdDM genes. To examine this, we analyzed the published layer-specific transcriptome data for *Arabidopsis* SAM cells³⁴ (Supplementary Fig. 8). This indicated that RdDM pathway genes showed no specific enrichment in any cell type, including L2. Thus, from the expression patterns of RdDM genes in L2, we could not conclude that L2

cell express a specific pattern of RdDM genes to establish a germ cell-specific methylation profile of the genome. In addition, unlike the *Arabidopsis* SAM, L2 cannot be clearly defined by cell lineage analysis in the SAM of grass species including rice³⁵. Further investigation will be required to reveal the cellular identity and contribution of subepidermal layer(s) of the SAM in grass species.”

2. Line 50, misspelling of 'begins'

Reply: We have corrected this error.

3. Line 92, Fig 1b It looks like there is a shift as well for CHG.

Reply: We agree with the reviewer’s comment, and have therefore described the shift for CHG methylation in lines 118 – 119 and 123 – 125 in the revised manuscript.

4. Line 97, Authors use the abbreviation TE (I assume transposable element) but I don't see it explicitly defined within the paper. It might make it more clear to use it together with the full name first.

Reply: We have explicitly defined the abbreviation TE as transposable element in the text where it appears first (line 73).

5. Line 87, Do authors have any theories why the CHH context was changed so dramatically relative to the CG or CHG contexts between the conditions examined? What is the possible underlying differences in functionality between these different types of methylation from previous studies that could explain this?

Reply: Our hypothesis is as follows for the reason why the CHH context was changed substantially relative to the CG or CHG context. In the vegetative-to-reproductive transition of the SAM, AGO4 protein level is high enough to be detectable by proteome analysis of the SAM (Fig. 4), and we think the increase in AGO4 protein contributes to the increase in CHH

hypermethylation. In contrast, for CG and CHG methylation pathway, the proteins functioning in this pathway cannot be detected by our proteome analyses. From this comparison, we think the differences in the protein amount contribute to differences in changes for CG/CHG and CHH contexts. The underlying difference in functionality of the methylation may be as follows. Generally, the functions of CG, CHG and CHH methylation are regulation of gene expression and silencing of TEs. The dramatic changes in CHH context may reflect an increase in the demand for TE silencing. In rice TEs, CHH methylation accumulates at the edges. This edge methylation is catalyzed by the activity of the RdDM pathway. RdDM pathway genes are highly expressed in the SAM, and 24-nt smRNA accumulates more in the SAM. The SAM requires stronger silencing activity for TEs to protect the genomic DNA of the stem cells and surrounding cells. This proposal is supported by the results showing higher CHH methylation in the vegetative SAM than in the leaf. In addition, the reproductive SAM requires further, stronger silencing for TEs than the vegetative SAM, since the former produces flowers and finally germ cells. We previously showed that the vegetative-to-reproductive transition in the SAM results in the silencing of TE expression (Tamaki et al., 2015), and so the increase in CHH methylation may contribute to this regulation.

These discussions are summarized in the following addition to the text (lines 262–264): “Taking these observations together, the higher accumulation of AGO4b in the SAM may explain why CHH methylation changes dramatically relative to CG or CHG methylation in the SAM ” and in the Discussion (lines 333 – 344): “Our findings suggest a novel and earlier link between DNA methylation changes in the SAM and germ cells. A recent methylome analysis of the male germ cell lineage in *Arabidopsis* showed that the reprogramming of DNA methylation occurs before meiosis³⁰. In the vegetative SAM, TEs are hypermethylated at their edges. This methylation is enforced in the vegetative-to-reproductive transition of the SAM. Germ cells differentiate in the reproductive SAM, and during this process many TEs are subjected to hypermethylation at their edges and in their bodies. We have shown that TE expression is reduced by the reproductive transition of the rice SAM²; hypermethylation may contribute to this silencing. Further hypermethylation in the CHH and CHG contexts in the germ cells may reflect a strict requirement for TE silencing in these cells. Such a two-step regulation may contribute to their attaining an epigenetic state suitable for them to function as germ cells.”

6. Line 104, For readers that do not follow methylation field closely, authors should consider elaborating on the biological significance of the differences observed for the CHG pattern given that the conclusions focus only on CHH? There appears to be a slight difference not just between the SAM and leaf but also between the reproductive and vegetative SAMs in Fig 2b, could, is there an explanation for this? Statistically one would expect some variation between all the groups for all the graphs in Fig2 and other graphs in this study. How did the authors draw the line between noise and true variation?

Reply: As the reviewer points out, CHG methylation has changed between the SAM and leaf and also between the vegetative SAM and reproductive SAM. This has been added to the text in lines 118 – 119 and 123 – 125 in the revised manuscript. Metaplots are suitable for presenting trends along the lengths of genomic features such as genes and TEs, and DMR analysis is used to statistically interpret changes in the methylation levels in specific regions.

7. Line 105, There also seem to be differences in the methylation of the upstream and downstream surroundings for TEs. in 2b not just at the edges.

Reply: We agree with the reviewer's comment and have added the following sentence at lines 140 -142: " CHH methylation is lowest in the mature leaf, high in the vegetative SAM, and highest in the reproductive SAM, in the close vicinity of genes and TEs, in the regions upstream and downstream of TEs, and at the edges of TEs."

8. Line 116, Primarily CHH hypermethylation might be a tool to control TE activity as you state later but could it possibly be multifunctional, since only a portion lines up with TEs?

Reply: As the reviewer suggests, CHH methylation is indeed known to have functions that are not limited to TE silencing, and some of the CHH hypermethylation found in this study may also have these additional functions. For example, CHH methylation at promoter regions can alter a gene's expression level. MITEs insert preferentially in the vicinity of genes (e.g., in gene promoters), and are highly methylated in the SAM, suggesting that they play a role in the

control of gene expression in the SAM. We have added this point to the Discussion (lines 327–331).

9. Line 118, Is this an overrepresentation of MITEs compared to their abundances in the entire TE population in the genome in general?

Reply: The response to this comment is the same as that to comment 6 raised by reviewer 1. We reiterate it here for convenience.

Reply: We performed additional analyses to determine the genome-wide profiles of different TE families in response to this comment. MITEs occupy 7% of the whole genome in length, but they occupy 41% of the total length of DMRs that are hypermethylated in the reproductive SAM relative to the vegetative SAM. This new result supported our previous conclusion and is shown in lines 158 – 161 and new Table 1. "Of the TE-overlapping Rep-hyperDMRs, 41.1% corresponded to miniature inverted-repeat transposable element (MITE)-type TEs (Fig. 3c) in nucleotide length, although MITEs comprise only 7.0% of the total length of TEs in the rice genome (Table 1)."

10. Line 125, Authors can just compare the inflorescence and leaf methylation patterns directly and come to a similar conclusion.

Reply: Thank you for the comment. In this analysis we aim to clarify whether CHH methylation varies among different developmental processes. To this end we compared Rep-hyper DMRs (hypermethylated in the reproductive SAM than in the vegetative SAM) and Leaf-hypo DMRs (hypermethylated in the vegetative SAM than in the leaf) . Both DMRs are based on the comparison of the vegetative SAM. This is because the vegetative SAM functions as a basis of the vegetative and reproductive development in plants; the vegetative SAM develops leaf in vegetative stage, and is converted to the reproductive SAM by vegetative-to-reproductive transition.

11. Line 152, Its pretty nice how well the Fig 4d RPM values and Fig2b avg methylation rate graphs line up with each other. There are tradeoffs with separate vs combined graph although it might be something to consider if you are looking to save space.

Reply: We appreciate the reviewer's comment, and have moved (former) Fig 4d to Fig 2b to line up with the methylation graphs.

12. Line 173, Authors tested overlap of *osdrm2* hypoDMRs with both RehyperDMR and LeafhypoDMR but they only seem to mention leaf results despite the former showing a similarly small overlap.

Reply: For the comparison between *osdrm2* DMRs and SAM DMRs, we refined our analysis using published methylome data (Tan et al., 2016) because the number and quality of their sequence reads are better than ours for *osdrm2* knockout plants. The data are presented in new Figure 4e, and we found that the majority of Rep-hyperDMRs, Leaf-hypoDMRs and Leaf-hyperDMRs are hypomethylated in *osdrm2*. The response to this comment is the same as a part of our response to a major comment raised by reviewer 1. We reiterate the corresponding part here for convenience.

Reply: In response to the reviewer's comment, we investigated the contribution of the RdDM pathway to the regions that show high CHH methylation in the SAM. According to the reviewer's suggestion, we tried to collect SAMs from our *osdrm2* knockout plants to conduct WGBS. However, *osdrm2* knockout plants show pleiotropic phenotypes including severe dwarfism, weakness, and sterility as described (Moritoh et al., 2012). Thus, we were unable to collect enough SAMs (more than 150) to obtain methylome data. We then re-examined our WGBS data for the *osdrm2* mutant. We found that the raw sequence data used in our previous manuscript were not of high quality and that the genome was insufficiently covered for robust analysis. We offer two reasons for this: 1) the methods used for library preparation, and the sequencing platform we used, were not up-to-date (the experiments were conducted in 2011), and 2) the number of PCR cycles required for library amplification were relatively high (18 cycles), thus decreasing the complexity of sequence reads, which in turn decreases coverage. We therefore tried to improve the analysis by replacing our own sequence data with new

sequence data that could be obtained from a public database. To this end, we searched publicly available data for WGBS of *osdrm2* mutants, which yielded the *osdrm2* WGBS data of Feng Tan et al. (2016). The number and quality of reads from these published data seemed better than ours. Using these data, we made density plots showing the frequency distribution of the CHH methylation differences among DMRs (new Figure 4e). In this analysis, we compared methylation differences between the *osdrm2* knockout leaf and the WT leaf. The regions analyzed were Rep-hyper CHH DMRs (i.e., CHH DMRs that are hypermethylated in the reproductive SAM), Leaf-hypo CHH DMRs (hypomethylated in leaf), and Leaf-hyper CHH DMRs (hypermethylated in leaf). The results indicated that the majority of these DMRs are hypomethylated in *osdrm2*. From these analyses, we concluded that OsDRM2 is the major contributor to CHH hypermethylation in the SAM during the floral transition. We have added this improved analysis to the manuscript. We appreciate the valuable comment from the reviewer.

Moritoh S, Eun CH, Ono A, Asao H, Okano Y, Yamaguchi K, Shimatani Z, Koizumi A, Terada R. (2012) Targeted disruption of an orthologue of DOMAINS REARRANGED METHYLASE 2, OsDRM2, impairs the growth of rice plants by abnormal DNA methylation. *Plant J.* 71:85-98.

Tan F., Zhou C., Zhou Q., Zhou S., Yang W., Zhao Y., Li G., Zhou D.-X. (2016) Analysis of chromatin regulators reveals specific features of rice DNA methylation pathways. *Plant Physiol.* 171: 2041-2054.

13. Do the authors think there is sufficient data to tie RdDM to being the primary cause of the methylation changes in the SAM, given the somewhat ambivalent evidence in Fig. 4f? while Fig. 3e also seems to point to a more complicated picture that cannot be linked to any one mechanism.

Reply: The response to this comment is the same as our response to comment 12 above. Using published sequencing data for the DNA methylome of *osdrm2*, we found that the majority of Rep-hyperDMRs, Leaf-hypoDMRs and Leaf-hyperDMRs are hypomethylated in *osdrm2*.

According to these analyses we concluded that OsDRM2 is the major contributor to CHH hypermethylation in the SAM during the floral transition.

14. Line 190, It could be helpful to have a graph directly combining and comparing the methylation rates of germ cells and the SAM.

Reply: In response to this comment, we arranged two meta-TE plots of germ cell methylation rate and SAM methylation rate next to each other in the new Figure 5b; the two plots were not superimposed on the same figure because the germ cell methylation rate and SAM methylation rate were measured in different experiments.

15. Line 190, The majority of TEs highly methylated in egg cells do not have hypermethylation in the SAM transition. Maybe there should be more explanation of what sets the hypermethylated egg TE set apart from the rest. Otherwise the overlap with the CHH floral hypermethylated TE set might seem a least partially by chance. In this context, linking these two temporally uncoupled (distant) events based on partial overlaps could be a mis or over interpretation.

Reply: In response to the reviewer's comment, we analyzed again the data concerning overlapping between the methylation in the SAM and in the germ cells. In addition to the egg cell methylome, we included sperm cell methylome data. Among the TEs highly methylated in both the egg cells and sperm cells for the CHH context, 62.9% of them have already undergone CHH hypermethylation in the SAM. The new results supported our previous conclusion, and are shown in a new figure (Fig. 5c).

16. Line 200, The conclusion might be expanded with a little more biological context/background to complement the observations. Authors must explain why MITEs are especially targeted for methylation based silencing of this mechanism?

Reply: We can think of three reasons why MITEs are targeted. The first is that MITEs are the most abundant TE in the rice genome, and thus targeting MITEs for methylation-based silencing can contribute to defending the genome during the reproductive transition. The second

is that MITEs tend to insert close to genes, whose expression can thus be affected. Silencing MITEs can contribute to effective regulation of the expression of genes with neighboring MITEs. The third reason is also related to the tendency of MITEs to be close to genes, namely that the RdDM machinery prefers to target euchromatic environments such as the vicinity of genes. These discussions are included in the following new text (lines 319–331):

" We found that MITEs are the preferred target of methylation in the SAM. The reason for this preference may be that MITEs are abundant in the rice genome and tend to insert near genes. TE silencing in the SAM may contribute to protecting the genome upon the onset of reproduction. To safeguard the genome against TE insertion, a reasonable strategy is to silence abundant TE families preferentially. MITEs are one of the most abundant TE families in rice and, because they tend to be inserted close to genes, their transcriptional activation may affect neighboring gene expression³³. A tendency to insert near genes is also observed for *Arabidopsis* and human TEs^{19–21}. For these reasons, silencing MITEs should be beneficial to protect the genome in the SAM. In addition to the silencing of TEs, CHH methylation at promoter regions can alter the gene expression level. MITEs prefer to insert into regions near genes such as promoters, and MITEs are highly methylated in the SAM, implying that they may often control gene expression in the SAM.."

17. Findings of the data linking these methylation events to a protective function by silencing TE activity during key events appears to be one of the central findings of the study but is only really brought up at the conclusion. Maybe this model could be introduced earlier/more prominently to allow readers to understand what idea is being reinforced.

Reply: We agree with the reviewer's comment, and have added text in the Introduction about this model as follows: "We ascribed this hypermethylation to a protective function for the genome by silencing TE activity."

Figures

Fig2a. There appears to be a slight difference in the methylation rates for CHG in the most highly methylated genes. Do authors have any theories on this? In some of the graphs there appears to be a central mass of green dots with few overlapping black dots and I'm not sure what the significance of that is or if that is actually a mass of both green and black dots that could be better visualized with a different color scheme.

Reply: In response to the reviewer's comment, we reorganized Figure 2 as boxplots for better visualization. Changes in CHG methylation are also reflected by the DMR analysis. We think this difference arises from changes in the chromatin environment between the SAM and the leaf.

Fig4b. Are the numbers across the top of the charts refer to replicate datasets?

Reply: These numbers indicate replicates of RNA-seq. We have clarified this in the revised legend of Figure 4b.

Fig. 5a. For metagene plots do the authors have any particular reason for using cutoffs of 3/4kb within the gene or is it just arbitrary?

Reply: The cutoff was chosen according to the report of Zemach et al., 2010; 3 kb upstream or downstream from gene is considered to include regulatory information for genes, and 4 kb from both the 5' and 3' end of a gene (8 kb in total) is considered sufficient to include the entire sequence of the majority of rice genes, whose average length is 3.2 kb.

Reference:

Zemach A, Kim MY, Silva P, Rodrigues JA, Dotson B, Brooks MD, Zilberman D (2010) Local DNA hypomethylation activates genes in rice endosperm. *Proc Natl Acad Sci USA*. 107:18729-18734.

Reviewer #4 (Remarks to the Author):

The study is dealing with DNA methylation in rice shoot apical meristem and proteomic analysis (subject of my review) forms only minor part of the whole study. The authors supported their conclusions based on DNA and RNA sequencing by proteomic analysis using two argonaut proteins OsAGO4a and OsAGO4b as examples. The applied proteomic approach and using emPAI for assessment of differences in protein levels within different tissues seems to be adequate for this purpose but I am missing detailed description of analytical procedures and data processing. I do not consider referring to previous publications satisfactory. I also miss basic parameters about identified proteins (score, number of peptides etc.). Please see below my detailed comments.

Supplementary material

Material and methods

Proteomic analysis

The description of proteomic analysis is very brief. The all procedure steps including data processing should be described in detail.

- how much input material was used (each contains 10 shoot apical meristems/replicate was mentioned only in reporting summary, how much leave material???)

We have added following description to Supplementary methods: “We collected 10 shoot apical meristems from each of 10 different plants for each replicate of the analysis. We also sampled and mixed three leaves from each of three different plants for each replicate.”

- add representative picture of SDS-PAGE separation of SAM/leave protein with marked region excised for analysis (or the whole lane was processed?)

We added SDS-PAGE of the SAM and leaf protein as Supplementary Figure 9. Whole lane was processed.

- digestion details (enzyme, time, temperature)

We have added the following descriptions to Supplementary methods.

“We extracted proteins from the samples with Laemmli sample buffer and incubated them at 65°C for 15 min, and then subjected them to SDS–PAGE [acrylamide concentration 10.5% (w/v)] (Supplementary Fig. 9). After staining by Flamingo (BioRad, Hercules, CA, USA), we sliced each lane of the gel into four parts of equal length. We washed the sliced pieces twice with HPLC-grade water with 30% (v/v) acetonitrile (Kanto Chemical, Tokyo, Japan), once with 100% acetonitrile, and then dried them in a vacuum concentrator. We treated the dried gel pieces with 2 µl of 0.5 µg µl⁻¹ trypsin (sequence grade; Promega, Madison, WI, USA) in 50 mM ammonium bicarbonate⁴⁴, followed by incubation at 37°C for 16 h. We recovered the digested peptides from the gel pieces by extracting twice with 20 µl of 5% (v/v) formic acid/50% (v/v) acetonitrile. The peptides were combined and dried in a vacuum concentrator.”

- description of LC-MS/MS analysis (I do not expect you used the exactly the same LC-MS/MS system and the same conditions as in ref 37 published in 2009)

We have added the following description to Supplementary methods.

“We performed LC-MS/MS analyses with an LTQ-Orbitrap XL-HTC-PAL-Paradigm MS4 system. Peptides were digested with trypsin and then loaded on the column (75 µm internal diameter, 15 cm; L-Column, CERI, Auburn, CA, USA). At this step we used a Paradigm MS4 HPLC pump (Michrom BioResources) and an HTC-PAL autosampler (CTC analytics, Zwingen, Switzerland). We prepared two buffers, 2% (v/v) acetonitrile and 0.1% (v/v) acetic acid (buffer A) and 90% (v/v) acetonitrile and 0.1% (v/v) acetic acid (buffer B). We applied a linear gradient from 5 to 45% by mixing buffers A and B, for 25 min, and after elution from the column we introduced the eluted peptides into an LTQ-Orbitrap mass spectrometer (Thermo Fisher Scientific, Bremen, Germany). The flow rate was 300 nl min⁻¹ and the spray voltage 2.0 kV. We set the range of MS scanning as m/z 200–2,000, and the top three peaks were subjected to MS/MS analysis.”

- conditions of database search – you really used exactly the same version of Mascot, the same mass accuracy, etc)

We have added the following description to Supplementary methods.

“We compared the spectra with the protein database OSMSU (<http://rice.plantbiology.msu.edu>) using the MASCOT server (version 2.4.1, Matrix Science, London, UK). We used the following parameters for the MASCOT search: set off the threshold at 0.05 in the ion score cut-off, peptide tolerance at 10 p.p.m., MS/MS tolerance at ± 0.5 Da, peptide charge of 2 + or 3 +, trypsin as enzyme allowing up to one missed cleavage, carbamidomethylation on cysteines as a fixed modification and oxidation on methionine as a variable modification.”

-it should be described how the normalized values of emPAI (or translational rates) were calculated (Fig. 4e and Suppl Table 4), the values to all proteins seems to me very low

We have added the following descriptions to Supplementary methods.

“We calculated the “normalized emPAI value” in Figure 4d by dividing emPAI of each protein by the sum of emPAI values of all the proteins in each sample^{45,46}. This normalization was applied to compare between different samples with different sums of emPAI values.”

Presentation of proteomic results

Suppl. Table 2

There are no details about quality of protein identification (score, number of peptides, sequence coverage) it should be added at least at the form as it presented in ref 37 – Suppl. Table 1.

We have revised Suppl. Table 1 by adding details about the quality of protein identification and presented as new Suppl. Table 3.

The proteins OsAGO4a and OsAGO4b is not possible to find in the table according their names used in the main text, there just Gene ID

We have revised Suppl. Table 1 by adding OsAGO4a and AGO4b to the table.

Suppl. Fig. 4

- the calculation of translational rates is not clear to me from the text "The translational rates were calculated by dividing the emPAI value into the relative expression level."

We defined translational rate as the ratio of protein accumulation (emPAI) to mRNA accumulation (RPKM). We have now corrected translational rates as emPAI/RPKM in Supplementary Figure 7.

Reviewers' comments:

Reviewer #1 (Remarks to the Author):

The revised version of the manuscript by Higo and co-workers has improved and most of my questions have been answered. The initial observation of increased CHH methylation in the shoot apical meristem (SAM) of rice is interesting, although given the recent methylome data from the Zilberman/Fischer/Scholten-lab indicates that this is only a first step of a more remarkable increase in RdDM (or other CHH-methylation) activity during reproduction, leading to strongly elevated levels of (CHH-)methylation in sperm and egg cells (Kim et al, PNAS 2019). The starting point of the current manuscript has been the observation that certain transposable elements (TEs) become stronger silenced during the transition from vegetative to reproductive SAM-development. If the increase in CHH methylation would account for TE silencing, did the authors test if TEs are further silenced in the egg cell and sperms? Using the same argument, did the authors test if the accumulation of CHH methylation over MITEs in proximity of genes correlates with reduced expression of these associated genes? I am not sure to which extent an increase in CHH methylation only would have such a strong effect, given that the methylation in other sequence contexts does not change. In that sense, the accumulation of sRNAs and CHH methylation would be simply a read-out of enhanced RdDM activity? And again, DNA methylation patterns might differ in the respective cell types of the SAM, not only the L2. For instance, differences in cell proliferation might account for DNA methylation differences, especially in the CHH context that requires the RdDM machinery and cannot rely on symmetry-based maintenance.

Further points are listed below:

Accession-numbers of NGS data are missing

Line 119: "slight shift of the peak" – precise the shift towards higher levels in the reproductive SAM.

Figure 2b: The shift in CHG is almost unchanged in vegetative vs. reproductive SAM, whereas CHH increases specifically in the latter. This was not discussed in the manuscript – do the authors have an explanation for this?

Suppl. Figure 2: Long TEs seem to gain specifically in CHH methylation, although short MITEs have been found to be the most highly enriched in CHH hypermethylation. Can the authors explain this effect?

Line 200-201: "CHH methylation affects TEs globally in different tissues." If I understood correctly, in the suppl. figure 5 the authors looked at MITEs, not all TEs? Please clarify/correct.

Line 216-217: "Asymmetric CHH sites need to be re-methylated after every cell division to maintain the methylated state." All methylated sites need to be remethylation after DNA replication. The difficulty with the asymmetric CHH context is that there is no symmetric hemimethylated double strand that can be used as a template. This needs to be rephrased.

Line 219-223: It seems the authors used published data in addition to their own RNA-seq data set for the transcriptome analysis (Figure 4b) – this should be stated/clarified.

Suppl. Figure 6: The increase in sRNAs in the reproductive SAM seems very limited or at least not especially elevated compared to the genome-wide meta-TE profile. I am not convinced that the CHH hyper-DMRs in the reproductive vs. vegetative SAM are explained by this subtle increase in sRNAs. To conclude this, one might need to analyze also the non-DMR TEs in comparison with the ones shown in the figure.

Figure 5C is missing, thought it contains important information!

Line 364-366: "DNA methylation in the sperm cell is regulated by mobile 21–22-nt smRNAs from its companion cell" is not correct. The cited paper described transcriptional silencing of the annotations targeted by sRNAs derived from the vegetative cell. In fact, RdDM seems to be downregulated in sperm in *A. thaliana* (Slotkin et al, Cell 2009). It has been proposed that linker histone H1 depletion might contribute to the elevated DNA methylation levels in sperm compared to other tissues (Hsieh et al, PNAS 2016)

However, there seems to be evidence that impairing active DNA demethylation in the vegetative cell in rice affects the methylome in the sperm cell in rice (Kim et al, 2019). The authors should cite this reference to make the point.

Reviewer #2 (Remarks to the Author):

The authors have now addressed all my concerns and performed additional analyses and improved the discussion.

Reviewer #4 (Remarks to the Author):

The authors answered all my raised questions and they corrected texts accordingly. The proteomic part of the work is acceptable.

I have just few moreless formal comments:

In supplementary notes

Line 94-97 in Accession codes section should be corrected „xxxx“ for correct numbers.

In new Suppl. Fig. 9 – SAM-R replicate 2 is twice and SAM-R replicate 2 is missing in the gel description.

Response to Comments from Reviewers

Reviewers' comments:

Reviewer #1 (Remarks to the Author):

The revised version of the manuscript by Higo and co-workers has improved and most of my questions have been answered. The initial observation of increased CHH methylation in the shoot apical meristem (SAM) of rice is interesting, although given the recent methylome data from the Zilberman/Fischer/Scholten-lab indicates that this is only a first step of a more remarkable increase in RdDM (or other CHH-methylation) activity during reproduction, leading to strongly elevated levels of (CHH-)methylation in sperm and egg cells (Kim et al, PNAS 2019).

The starting point of the current manuscript has been the observation that certain transposable elements (TEs) become stronger silenced during the transition from vegetative to reproductive SAM-development. If the increase in CHH methylation would account for TE silencing, did the authors test if TEs are further silenced in the egg cell and sperms?

Reply: In response to the reviewer's comment, we tested whether TEs are further silenced in the egg cell and sperm cell. We categorized expressed TEs into four groups: (1) those showing higher CHH methylation in reproductive SAM than in vegetative SAM, (2) those showing lower CHH methylation in reproductive SAM than in vegetative SAM, (3) those with similar CHH methylation in both SAM and (4) those without detectable CHH methylation in either SAM. We then compared expression levels of TEs in vegetative SAM (Hd3a RNAi, Tamaki et al., PNAS

2015), reproductive SAM (Tamaki et al., PNAS 2015), and egg cell and sperm cell (Anderson et al., 2013) for each group of TEs (new Supplementary Fig. 11). We could not find any relationship between expression level and TE group, suggesting that SAM-expressed TEs are not further silenced in egg or sperm cells. We then thought about the role of CHH methylation in the SAM, and considered that it might be to reinforce silencing of "silenced TEs". We should note that the above analysis focused on expressed TEs that were not silenced enough to be suppressed in the SAM. The majority of TEs are already methylated and silenced in the SAM. Silencing of those TEs may be reinforced through an increase in CHH methylation in the reproductive SAM and in germ cells. Interestingly, through the analysis shown in new Supplementary Fig. 11, we found that TEs expressed and methylated in the SAM (group (1) and (2)) tend to be silenced in the egg cell, whereas TEs expressed but unmethylated in the SAM (group (4)) tend to be expressed, or to escape from silencing, in the egg cell. From this observation, we speculate that CHH methylation in the SAM distinguishes TEs to be silenced in the egg cell.

We appreciate these noteworthy and insightful comments from reviewer 1. We have summarized the above by adding the following new text (lines 327–343): “To examine whether TEs are further silenced in the egg cell and sperm cell, we analyzed expression levels of TEs methylated in the SAM. We categorized expressed TEs into four groups: (1) those showing higher CHH methylation in reproductive SAM than in vegetative SAM, (2) those showing lower CHH methylation in reproductive SAM than in vegetative SAM, (3) those with similar CHH methylation in both SAMs, and (4) those without detectable CHH methylation in either SAM. We then compared expression levels of TEs in vegetative SAM, reproductive SAM, egg cell³², and sperm cell³² for each TE group (Supplementary Fig. 11). We found no relationship between expression level and TE group, suggesting that SAM-expressed TEs are not further silenced in egg or sperm cells. Although expressed TEs were evidently not sufficiently silenced for their expression in the SAM to be prevented, the majority of TEs are already methylated and silenced in the SAM. Silencing of these TEs may be reinforced through an increase in CHH methylation

in the reproductive SAM and in germ cells.

Interestingly, we found that TEs expressed and methylated in the SAM (group (1) and (2)) tend to be silenced in the egg cell, whereas TEs expressed but unmethylated in the SAM (group (4)) tend to be expressed, or to escape from silencing, in the egg cell (Supplementary Figs. 11 and 12). From this observation, we speculate that CHH methylation in the SAM may distinguish TEs that are to be silenced in the egg cell.”

Using the same argument, did the authors test if the accumulation of CHH methylation over MITEs in proximity of genes correlates with reduced expression of these associated genes? I am not sure to which extent an increase in CHH methylation only would have such a strong effect, given that the methylation in other sequence contexts does not change. In that sense, the accumulation of sRNAs and CHH methylation would be simply a read-out of enhanced RdDM activity?

Reply: According to the reviewer's comment, we tested whether accumulation of CHH methylation over MITEs situated in proximity to genes correlates with reduced expression of these nearby genes. We analyzed CHH methylation levels of MITEs in proximity to differentially expressed genes in the vegetative-to-reproductive transition in the SAM (Tamaki et al., PNAS 2015). The results indicated that changes in the gene expression levels do not correlate with changes in the CHH methylation levels (new Supplementary Fig. 8). Thus, changes in CHH methylation may not have a strong effect on proximal gene expression in general. The accumulation of CHH methylation would be a read-out of enhanced RdDM activity. The above considerations are summarized as follows and inserted into the Results (lines 229–234): “To examine whether accumulation of CHH methylation over MITEs in proximity to genes affects the expression of these nearby genes, we analyzed CHH methylation levels of MITEs in proximity to genes that are differentially expressed in the vegetative-to-reproductive transition in

the SAM. We found that the changes in the gene expression levels did not correlate with changes in the CHH methylation levels (Supplementary Fig. 8), suggesting that CHH methylation does not have a strong effect on proximal gene expression in the SAM.”

And again, DNA methylation patterns might differ in the respective cell types of the SAM, not only the L2. For instance, differences in cell proliferation might account for DNAmethylation differences, especially in the CHH context that requires the RdDM machinery and cannot rely on symmetry-based maintenance.

Reply: We thank the reviewer for this insightful remark on the cell type specificity of DNA methylation in the SAM. As the reviewer points out, CHH methylation can be more sensitive to cell proliferation activity, because it depends on the RdDM machinery and is not supported by symmetry-based maintenance. These considerations are summarized as follows and inserted into the Discussion (lines 413–417): “However, it is still possible that DNA methylation levels differ among cell types. Different cell types in the SAM show different cell proliferation activity, which may affect the cellular specificity of DNA methylation level: CHH methylation may well be more sensitive to cell proliferation activity, because it depends on the RdDM machinery and is not supported by symmetry-based maintenance.”

Further points are listed below:

Accession-numbers of NGS data are missing

Reply: We have corrected the accession code. The code is DRA007588.

Line 119: “slight shift of the peak” – precise the shift towards higher levels in the

reproductive SAM.

Reply: We have changed this sentence to make the description precise (lines 114–115): “For CHG methylation, we observed a slight shift of the peak toward higher levels in the reproductive SAM (Fig. 1b).”

Figure 2b: The shift in CHG is almost unchanged in vegetative vs. reproductive SAM, whereas CHH increases specifically in the latter. This was not discussed in the manuscript – do the authors have an explanation for this?

Reply: The question of why CHH methylation increases but CHG methylation does not in the reproductive SAM is an interesting point, but the reason is unclear. It may reflect a difference in activity between pathways responsible for CHH methylation and CHG methylation. For CHH methylation, we showed that RdDM is activated in the reproductive SAM, probably through an increase in OsAGO4b protein. In contrast, for CHG methylation, CMT3 mRNA level did not change during the vegetative-to-reproductive transition of the SAM according to our RNA-seq analysis, and CMT3 protein level was below the detection limit of our proteome analysis. However, Tang et al. (Plant Physiol. 2016) showed that CHG methylation, along with CHH methylation, is reduced in the *osdrm2* mutant of rice, suggesting that OsDRM2 and RdDM pathway affect CHG methylation. It is possible that CHG methylation changes with an increase in CHH methylation, but the maintenance of CHG methylation is not effective in the SAM. Actually, it has been observed that CHH methylation changed whereas CHG methylation unchanged in the context of seed development and root cell type specificity in *Arabidopsis* (Kawakatsu et al. 2017 Nature Plants, Bouyer et al. 2017 Genome Biol, Lin et al. 2017 PNAS, Narsai et al. 2017 Genome Biol, Kawakatsu et al. 2017 Genome Biol). These comments are summarized as follows and inserted into the Discussion (lines 357–369): “DNA methylation in

the SAM is regulated through context-specific mechanisms. In our analysis, CHG methylation is almost unchanged, but CHH methylation increases, in the vegetative-to-reproductive transition of the SAM (Fig. 2b). A possible reason for this is the difference in activity between pathways responsible for CHH methylation and CHG methylation: RdDM is activated in the reproductive SAM through OsAGO4b protein accumulation (Fig. 4), but the mRNA level of *CMT3*, an enzyme responsible for CHG methylation in plants, is unchanged during the vegetative-to-reproductive transition in the SAM and CMT3 protein is below the detection limit of our proteome analysis. However, there may be a link between CHG methylation and CHH methylation because the *osdrm2* mutant shows a reduction of CHG methylation along with a reduction of CHH methylation. It has been observed that CHH methylation changes whereas CHG methylation is unchanged in the context of seed development and root cell-type specificity in *Arabidopsis*^{9–12,33}. Further study is needed to clarify the context specificity of methylation regulation in the SAM.”

Suppl. Figure 2: Long TEs seem to gain specifically in CHH methylation, although short MITEs have been found to be the most highly enriched in CHH hypermethylation. Can the authors explain this effect?

Reply: Supplementary Fig. 2 shows that both short and long TEs gain CHH methylation, but that the CHH methylation rate is higher in short TEs than in long TEs: CHH methylation rates are 10–25% for short TEs and 5–18% for long TEs. The higher methylation rate in short TEs enabled us to detect DMRs more effectively in those elements than in longer TEs, resulting in the higher enrichment of short TEs, especially MITEs, among DMR-overlapping TEs in Figure 2b. This is summarized with the addition of the following text to the main text (lines 155–160): “Both short and long TEs gain CHH methylation (Supplementary Fig. 2), although short MITEs are most highly enriched for CHH methylation (Fig. 3c). The reason for this difference may be

that methylation rates vary according to TE length. CHH methylation rates are higher in short TEs (10–25% methylated at peaks) than in long TEs (5–18%) (Supplementary Fig. 2), resulting in effective DMR detection that leads to enrichment in short TEs, especially MITEs, among DMR-overlapping TEs (Fig. 3c).”

Line 200-201: “CHH methylation affects TEs globally in different tissues.” If I understood correctly, in the suppl. figure 5 the authors looked at MITEs, not all TEs? Please clarify/correct.

Reply: Yes, we looked at MITEs in Supplementary Fig. 5, and thus corrected this sentence in the main text (lines 195 - 196): “CHH methylation affects MITEs globally in different tissues.”

Line 216-217: “Asymmetric CHH sites need to be re-methylated after every cell division to maintain the methylated state.” All methylated sites need to be remethylation after DNA replication. The difficulty with the asymmetric CHH context is that there is no symmetric hemimethylated double strand that can be used as a template. This needs to be rephrased.

Reply: We agree with the reviewer’s comment, and have therefore rephrased this sentence in the main text (lines 242–245): “All the methylation sites need to be re-methylated after every DNA replication to maintain the methylation rate. In asymmetric CHH sites, there is no symmetric hemimethylated double strand that can be used as a template for remethylation.”

Line 219-223: It seems the authors used published data in addition to their own RNA-seq data set for the transcriptome analysis (Figure 4b) – this should be stated/clarified.

Reply: RNA-seq datasets for the transcriptome analysis in Figure 4b consist of the datasets from

our previous publication (Tamaki et al. (2015) PNAS) and our new datasets: Three replicates of the reproductive SAM and replicate 1 in the leaf are from Tamaki et al. (2015) PNAS, and the remaining replicates of the reproductive SAM and leaf and all eight replicates of the vegetative SAM are our new datasets. This is stated to clarify the origin of the datasets in the main text (lines 246–250): “To determine whether genes for CHH methylation pathways are expressed in the SAM, we conducted RNA-seq of the SAM. This dataset was combined with the RNA-seq dataset for the SAM and leaf from our previous publication²: replicate 1 in the leaf and replicates 1–3 in the reproductive SAM are from our previous publication² and other RNA-seq datasets are from our new experiments.”

Suppl. Figure 6: The increase in sRNAs in the reproductive SAM seems very limited or at least not especially elevated compared to the genome-wide meta-TE profile. I am not convinced that the CHH hyper-DMRs in the reproductive vs. vegetative SAM are explained by this subtle increase in sRNAs. To conclude this, one might need to analyze also the non-DMR TEs in comparison with the ones shown in the figure.

Reply: We agree with the reviewer's comment: the increase in smRNAs in the reproductive SAM is limited. In addition, following the reviewer's suggestion, we also created a metaplot of smRNAs for non-DMR TEs. We found that smRNAs were higher in TEs overlapping with CHH hyper-DMRs than in non-DMR TEs. The metaplot indicated that for non-DMR TEs, smRNAs were high in the reproductive stage, but there was only a small difference. Thus, in the new text, we concluded that the contribution of smRNA increase to the CHH methylation increase is limited. The new metaplot was added to the new Supplementary Fig. 9, with addition of the following sentences in the main text (lines 273–280): “To assess this more closely we analyzed smRNA profiles along the TEs overlapping with Rep-hyper DMRs and compared them with the smRNA profiles of the TEs not overlapping with those DMRs. This revealed that smRNAs were

more abundant in TEs overlapping with Rep-hyper DMRs than in TEs not overlapping with the DMRs, whereas smRNAs were increased slightly in the reproductive SAM at the edges of TEs both overlapping and non-overlapping with the DMRs (Supplementary Fig. 9). Together, these results imply that 24-nt smRNA abundance in the SAM makes a limited contribution to the increase in CHH hypermethylation in the reproductive SAM.”

Figure 5C is missing, thought it contains important information!

Reply: We have explained about Figure 5c and these sentences are highlighted in the new main text (lines 323–326): “To test this, we asked whether TEs that are hypermethylated in germ cells include those that are hypermethylated in the SAM. Among the TEs highly methylated in both the egg cells and sperm cells for the CHH context, 46.3% had already undergone CHH hypermethylation in the SAM (Fig. 5c, $p < 2.2e-16$; Fisher’s exact test).”

Line 364-366: “DNA methylation in the sperm cell is regulated by mobile 21–22-nt smRNAs from its companion cell” is not correct. The cited paper described transcriptional silencing of the annotations targeted by sRNAs derived from the vegetative cell. In fact, RdDM seems to be downregulated in sperm in *A. thaliana* (Slotkin et al, Cell 2009). It has been proposed that linker histone H1 depletion might contribute to the elevated DNA methylation levels in sperm compared to other tissues (Hsieh et al, PNAS 2016) However, there seems to be evidence that impairing active DNA demethylation in the vegetative cell in rice affects the methylome in the sperm cell in rice (Kim et al, 2019). The authors should cite this reference to make the point.

Reply: We thank the reviewer for these noteworthy and insightful remarks. As the reviewer points out, the cited paper did not describe DNA methylation in the sperm cell as being regulated

by mobile 21–22-nt smRNAs from its companion cell: rather, the paper described the silencing of annotations, or transcription units, targeted by smRNA from the vegetative cells. As the reviewer notes, Slotkin et al. (2009) indicated that RdDM seems to be downregulated in the sperm cell, and Hsieh et al. (2016) reported that depletion of linker histone H1 might be involved in the increase in DNA methylation in the sperm cell in *Arabidopsis*. These papers suggest that hypermethylation in the sperm cell is achieved through a mechanism involving depletion of linker histone H1, and RdDM by mobile smRNA derived from the vegetative cell is not supported. However, Kim et al. (2019) find that the sperm cell methylome is promoted indirectly by DNA demethylation in the vegetative cell. These points are summarized in the following addition to the main text (lines 423–429): “In *Arabidopsis* pollen, transcriptional silencing has been reported for annotations that are targeted by smRNAs derived from the vegetative cell^{7,8,29}. Whether hypermethylation by mobile smRNAs affects this silencing is debated. Hypermethylation in sperm cells may be caused by the depletion of linker histone H1 in *Arabidopsis*⁴⁰, and the contribution of RdDM may be limited because RdDM seems to be downregulated in the sperm cell⁸. However, Kim et al.³¹ found that the sperm cell methylome is promoted indirectly by DNA demethylation in vegetative cell in rice.”

Reviewer #2 (Remarks to the Author):

The authors have now addressed all my concerns and performed additional analyses and improved the discussion.

Reply: We are pleased to hear that the reviewer is satisfied with our revisions to the manuscript.

Reviewer #3 (Remarks to the Author):

Unfortunately, we were unable to receive further comment from reviewer #3. We asked reviewer #1 to comment on your response to this reviewer. Reviewer #1 suggested that the

following should be considered:

1. In order to support the claim that increased CHH DNA methylation may contribute to TE silencing during the transition from vegetative to reproductive development, please compare TEs that gain CHH methylation to those that are repressed in the reproductive meristem. i.e. whether the fraction of expressed TEs in the SAM that become repressed during the transition to reproductive phase correspond to those that gain methylation in the reproductive meristem compared to the vegetative meristem. As CHG methylation also increases and as CG methylation is already prominent, it would appear that DNA methylation increases globally in the reproductive meristem, possibly due to RdDM.

Reply: We thank the reviewer for this insightful remark. In response, we analyzed whether the TEs whose expression is repressed in the reproductive SAM (SAM-R) (repressed TEs) correspond to those that gain CHH methylation in the SAM-R, and the results are shown in new Supplementary Fig. 6. Our previously published transcriptome data identified 8,045 expressed TEs: of these, 4,263 TEs gained CHH methylation in the SAM-R. In contrast, those previous data identified 526 repressed TEs: of these, 292 TEs gained CHH methylation. This comparison suggests that half of the repressed TEs are regulated by CHH hypermethylation, whereas the remaining half are regulated by an unknown mechanism. We also analyzed the ratio of CHH methylation in the vegetative SAM (SAM-V) to that in the SAM-R for expressed TEs and repressed TEs by scatter plot. The plot showed that both types of TEs gained CHH methylation similarly. These results suggest that DNA methylation increases globally in the reproductive meristem, possibly due to the activation of RdDM. We think that this global increase in CHH methylation may consolidate the silencing of TEs that are not expressed even in the SAM, and/or may be a prerequisite for repressing a subset of TEs prior to reproduction. These comments are summarized in the following addition to the main text with new Supplementary Fig. 6 (lines

203–215): “To examine whether increased CHH methylation contributes to TE silencing during the transition from vegetative to reproductive transition, we compared TEs that gain CHH methylation and those that are repressed in the reproductive SAM. From our previously published data, we identified 8,045 expressed TEs in the SAM; of these, 4,263 gained CHH methylation in the reproductive SAM. In contrast, we identified 526 repressed TEs, of which 292 gained CHH methylation (Supplementary Fig. 6a). This suggests that repressed TEs were not necessarily enriched for CHH-hypermethylated TEs. We also analyzed the ratio of CHH methylation in the vegetative SAM and reproductive SAM for expressed and repressed TEs by scatter plot (Supplementary Fig. 6b), which showed that both types of TEs gained CHH methylation similarly. These results suggest that DNA methylation increases globally in the reproductive SAM, possibly due to the activation of RdDM (discussed later). This global increase in CHH methylation may consolidate silencing of TEs that are not expressed even in the SAM, and/or be a prerequisite for repressing a subset of TEs prior to reproduction.”

We also revised Discussion as follows (lines 371–378): “TEs are highly expressed in the vegetative SAM, but some TEs are silenced during the vegetative-to-reproductive transition². We asked whether this silencing correlates with the increase in CHH methylation in the reproductive SAM (Supplementary Fig. 6), and found that half of the expressed TEs are hypermethylated, and half of them are silenced. This suggests that silencing of expressed TEs does not necessarily coincide with CHH hypermethylation. We speculate that the function of CHH hypermethylation in the SAM is to reinforce pre-existing silencing of TEs. In the *Arabidopsis* SAM, TE silencing appears to be enforced via high expression of RdDM pathway components³⁴, and our findings thus illuminate a conserved feature in flowering plants.”

2. To further explore whether CHH methylation contributes to regulation of gene expression, previously published transcriptome data (Tamaki et al. 2015 PNAS) could be analysed.

Reply: Following the reviewer's comment, we reanalyzed our previously published transcriptome data to determine whether CHH methylation contributes to the regulation of gene expression. Based on Tamaki et al. (2015) PNAS, we classified all genes in rice into genes whose expression is activated, repressed, or otherwise expressed in the SAM. Next, we plotted CHH methylation in the SAM-V and SAM-R for genes in each classification, divided into 1 kb upstream of the gene to the start of the annotation, the gene body, and 1 kb downstream of the end of the annotation. The results showed that the CHH methylation of the SAM-R was higher than that of the SAM-V in the upstream and downstream regions of all the classifications. This suggests that CHH methylation is also globally up-regulated in the genes during the transition from SAM-V to SAM-R. A closer inspection of the plots revealed that genes whose expression is repressed are characterized by CHH methylation of the gene body already being high at the SAM-V stage. These points are summarized in the following addition to the main text with new Supplementary Fig. 7 (lines 217–227): “To further explore whether CHH methylation contributes to regulation of gene expression, we analyzed previously published transcriptome data. All rice genes were categorized as genes whose expression is activated, repressed, or otherwise expressed in the SAM. We then plotted CHH methylation in the vegetative and reproductive SAM for the genes in each classification, divided into 1 kb upstream of the start of annotation, the gene body, and 1 kb downstream of the end of annotation (Supplementary Fig. 7). The results showed that CHH methylation in the reproductive SAM was higher than that in the vegetative SAM in the upstream and downstream regions of all groups. This suggests that CHH methylation is also globally up-regulated in genes during the transition from vegetative to reproductive SAM. We also found that genes whose expression is repressed are characterized by CHH methylation of the gene body already being high at the vegetative stage.”

3. A statistical test should be performed to assess the significance of the overlap between

TEs showing an increase in CHH methylation during vegetative to reproductive transition and TEs that are CHH hypermethylated in the egg cell (Fig. 5c)

Reply: We performed Fisher's exact test to assess the significance of the overlap between high CHH methylation in egg and sperm cells and high methylation in the SAM among TEs. The result is $p < 2.2e-16$, and thus the overlap between SAM methylation and germ cell methylation is significant. During this analysis, we discovered that we had duplicated gene counts in the previous Fig. 5c, and so we removed the duplicates and revised this figure. In the new Fig. 5c, of the TEs that were methylated in the egg or sperm cell, 46.3% were also methylated in the SAM. The above statistical test was performed on the basis of this new exact value. This information was added to the main text (lines 323–326), together with new Fig. 5c.

Reviewer #4 (Remarks to the Author):

The authors answered all my raised questions and they corrected texts accordingly. The proteomic part of the work is acceptable.

I have just few moreless formal comments:

In supplementary notes

Line 94-97 in Accession codes section should be corrected „xxxx“ for correct numbers.

Reply: We have corrected the accession code. The code is DRA007588.

In new Suppl. Fig. 9 – SAM-R replicate 2 is twice and SAM-R replicate 2 is missing in the gel description.

Reply: We have corrected this mis-labelling in response to the reviewer's comment.

REVIEWERS' COMMENTS:

Reviewer #1 (Remarks to the Author):

The second revision of the manuscript by Higo and co-workers has improved significantly and all my remarks have been addressed properly. There are only few points that need to be clarified before publication:

1) Figure 3b & c, Suppl. Figure 3a & b : Include the genome-wide distribution of TE classes and families.

2) Line 207-208 and suppl. Figure S6a: The Venn diagram states "TEs expressed in SAM-V and SAM-R", but within this group there are TEs named "TEs repressed in SAM-R". This is somewhat contradictory; do these TEs show lower expression level in the reproductive SAM compared to the vegetative SAM? These needs to be clarified.

3) Line 219: "() genes whose expression is activated, repressed, or otherwise expressed ()". What is meant by 'otherwise expressed'? Constitutively expressed? Please correct/explain.

4) Line 225-227: "We also found that genes whose expression is repressed are characterized by CHH methylation of the gene body already being high at the vegetative stage." I guess this is about TEs that become repressed in the reproductive SAM? Please clarify/correct.

5) Line 257-258: "hyper-accumulation of 24-nt smRNAs contributes to CHH hypermethylation". Please change 'hyper-accumulation' to 'accumulation' or 'elevated levels'.

6) Suppl. Figure 10: The title of the figure mentions smRNAs, but there are no data shown with respect to smRNA abundance. Please correct.

7) Line 327-343: The results suggest that TEs become silenced during the transition from vegetative to reproductive growth, independently of the change in CHH methylation. In addition TEs that carry CHH methylation are generally less expressed than those are devoid of CHH methylation. I would restrict the interpretation to

8) Line 344-345: "These results imply that a proportion of the TEs experience CHH hypermethylation in the SAM before being subjected to hypermethylation in the germ cells." As the term 'hyper' is always used relative to another stage, an alternative expression should be used here, such as 'elevated levels of CHH methylation in the SAM that further increase in the germ cells'.

9) Line 380: "MITEs are the preferred target of methylation in the SAM". I guess the authors mean CHH methylation?

10) Line 405-406: "L2 cells express RdDM pathway genes to confer a CHH methylation pattern similar to that in the germ cells". It should be kept in mind that RdDM is diminished in pollen sperm cell, at least in Arabidopsis (Slotkin et al, Cell 2009). So this sentence would make sense for the egg cell, but less so for the sperm cell.

11) Line 423-425: "In Arabidopsis pollen, transcriptional silencing has been reported for annotations that are targeted by smRNAs derived from the vegetative cell. Whether hypermethylation by mobile smRNAs affects this silencing is debated." It should be kept in mind that the RdDM machinery is downregulated in sperm cell in Arabidopsis pollen (see above). It is more likely that these transposable elements are post-transcriptionally silenced.

12) Line 428-429: "() the sperm cell methylome is promoted indirectly by DNA demethylation in vegetative cell in rice." This needs to be corrected. In fact, the loss of active DNA demethylation in pollen vegetative cells leads to a decrease of CHH methylation in sperm cells as shown in the cited publication.

Response to comments from Reviewers

REVIEWERS' COMMENTS:

Reviewer #1 (Remarks to the Author):

The second revision of the manuscript by Higo and co-workers has improved significantly and all my remarks have been addressed properly. There are only few points that need to be clarified before publication:

1) Figure 3b & c, Suppl. Figure 3a & b : Include the genome-wide distribution of TE classes and families.

Reply: According to the reviewer's comment, we included genome-wide distribution of TE classes and families in the revised Supplementary Figure 3a and 3b.

2) Line 207-208 and suppl. Figure S6a: The Venn diagram states "TEs expressed in SAM-V and SAM-R", but within this group there are TEs named "TEs repressed in SAM-R". This is somewhat contradictory; do these TEs show lower expression level in the reproductive SAM compared to the vegetative SAM? These needs to be clarified.

Reply: Thank you for pointing this. We have changed this indication to make the description precise in revised Supplementary Figure 6a: "TEs expressed in SAM-V and SAM-R" are changed to "TEs that show lower expression level in SAM-R compared to SAM-V".

3) Line 219: "() genes whose expression is activated, repressed, or otherwise expressed ()". What is meant by 'otherwise expressed'? Constitutively expressed? Please correct/explain.

Reply: The genes categorized to those “otherwise expressed” indicate constitutively expressed genes in the SAM. We have corrected this in the manuscript.

4) Line 225-227: "We also found that genes whose expression is repressed are characterized by CHH methylation of the gene body already being high at the vegetative stage." I guess this is about TEs that become repressed in the reproductive SAM? Please clarify/correct.

Reply: Supplementary Figure 7 focuses on CHH methylation in genes: this analysis was intended to explore whether CHH methylation contributes to regulation of gene expression.

5) Line 257-258: "hyper-accumulation of 24-nt smRNAs contributes to CHH hypermethylation". Please change 'hyper-accumulation' to 'accumulation' or 'elevated levels'.

Reply: We have corrected to the word 'hyper-accumulation' to 'elevated levels'.

6) Suppl. Figure 10: The title of the figure mentions smRNAs, but there are no data shown with respect to smRNA abundance. Please correct.

Reply: We have corrected the title of Supplementary Fig. 10 as “Protein levels in vegetative SAM and reproductive SAM.”.

7) Line 327-343: The results suggest that TEs become silenced during the transition from vegetative to reproductive growth, independently of the change in CHH methylation. In addition TEs that carry CHH methylation are generally less expressed than those are devoid of CHH methylation. I would restrict the interpretation to

Reply: In response to the reviewer's comment, we have added the following sentence "We found no relationship between expression level and TE group, suggesting that TEs become silenced during the transition from vegetative to reproductive stages, independently of the change in CHH methylation and that SAM-expressed TEs are not further silenced in egg or sperm cells."

8) Line 344-345: "These results imply that a proportion of the TEs experience CHH hypermethylation in the SAM before being subjected to hypermethylation in the germ cells." As the term 'hyper' is always used relative to another stage, an alternative expression should be used here, such as 'elevated levels of CHH methylation in the SAM that further increase in the germ cells'.

Reply: We agree with the reviewer's comment, and have therefore changed this sentence to "These results imply that a proportion of the TEs experience elevated levels of CHH methylation in the SAM that further increase in the germ cells".

9) Line 380: "MITEs are the preferred target of methylation in the SAM". I guess the authors mean CHH methylation?

Reply: As the reviewer pointed, we meant that MITEs are target for CHH methylation. We therefore changed this sentence "We found that MITEs are the preferred target of CHH methylation in the SAM".

10) Line 405-406: "L2 cells express RdDM pathway genes to confer a CHH methylation pattern similar to that in the germ cells". It should be kept in mind that RdDM is diminished in pollen sperm cell, at least in Arabidopsis (Slotkin et al, Cell 2009). So this sentence would make sense for the egg cell, but less so for the sperm cell.

Reply: Thank you for pointing this. We have revised this sentence by mentioning that this makes sense for the egg cell but less so for the sperm cell. “L2 cells express RdDM pathway genes to confer a CHH methylation pattern similar to that in the germ cell, especially for egg cells: this is not so evident for the sperm cells because RdDM is diminished in pollen sperm cell⁸”

11) Line 423-425: "In Arabidopsis pollen, transcriptional silencing has been reported for annotations that are targeted by smRNAs derived from the vegetative cell. Whether hypermethylation by mobile smRNAs affects this silencing is debated."

It should be kept in mind that the RdDM machinery is downregulated in sperm cell in Arabidopsis pollen (see above). It is more likely that these transposable elements are post-transcriptionally silenced.

Reply: We appreciate the reviewer’s comment. We included the perspective from the post-transcriptional silencing in this sentence. “In Arabidopsis pollen, transcriptional silencing has been reported for annotations that are targeted by smRNAs derived from the vegetative cell. It is likely that these TEs are post-transcriptionally silenced. Whether hypermethylation by mobile smRNAs affects this silencing is debated.”

12) Line 428-429: "() the sperm cell methylome is promoted indirectly by DNA demethylation in vegetative cell in rice." This needs to be corrected. In fact, the loss of active DNA demethylation in pollen vegetative cells leads to a decrease of CHH methylation in sperm cells as shown in the cited publication.

Reply: Thank you for this comment. We have corrected by changing this sentence as follows: “the loss of active DNA demethylation in pollen vegetative cells leads to a decrease of CHH methylation in sperm cells in rice”.